# SELECTIVE STATE-SPACE MODELING OF CORRELATION MAPS FOR SEMANTIC CORRESPONDENCE

## ABSTRACT

Establishing semantic correspondences between images is a fundamental yet challenging task in computer vision. Traditional feature-metric methods enhance visual features but may miss complex inter-image relationships, while recent correlation-metric approaches attempt to model these relationships but are hindered by high computational costs due to processing 4D correlation maps. We introduce MambaMatcher, a novel method that overcomes these limitations by efficiently modeling high-dimensional correlations using selective state-space models (SSMs). By implementing a similarity-aware selective scan mechanism adapted from Mamba's linear-complexity algorithm, MambaMatcher refines the 4D correlation tensor effectively without compromising feature map resolution or receptive field. Experiments on standard semantic correspondence benchmarks demonstrate that MambaMatcher achieves state-of-the-art performance without relying on large input images or computationally expensive diffusion-based feature extractors, effectively capturing rich inter-image correlations while maintaining computational efficiency.

## 1 INTRODUCTION

Establishing semantic correspondences between images is a fundamental problem in computer vision, with wide-ranging applications in augmented and virtual reality (AR/VR), such as virtual try-on (Kim et al., 2023), edit propagation (Endo et al., 2016; Peebles et al., 2022), and instance swapping (Zhang et al., 2024). The task involves identifying semantically corresponding regions between pairs of images depicting different instances of the same class (Cho et al., 2015; Min et al., 2019; Truong et al., 2022). Despite significant advancements in deep learning, reliably establishing semantic correspondences remains challenging due to substantial intra-class variations, including differences in pose, scale, and appearance among instances.

Current state-of-the-art methods predominantly adopt a *feature-metric approach*, enhancing the quality of features at each spatial position in the images. This enhancement is achieved by either (i) employing more powerful feature extractors (Tang et al., 2023; Zhang et al., 2024) or (ii) improving feature representations through aggregation with additional convolutional or attentional layers (Seo et al., 2018; Lee et al., 2021b; Luo et al., 2024). While richer features can robustly identify semantic similarities across local pixels, feature-metric methods may struggle to capture complex inter-image relationships due to their focus on individual feature points.

An alternative is the *correlation-metric approach*, where methods aim to model inter-image relationships by processing the 4D correlation map between features from the two images (Rocco et al., 2018; Min & Cho, 2021; Cho et al., 2021; Kim et al., 2022). Although this approach can alleviate ambiguous or noisy correspondences by considering global correlation patterns, it suffers from significant computational complexity. Processing the 4D correlation map incurs up to quartic complexity with respect to the feature map dimensions, which severely limits the feature map resolution and necessitates compromises on the receptive field or network expressivity—critical factors for accurate and robust correspondences. Consequently, despite their potential, correlation-metric methods are often outperformed by feature-metric methods that utilize stronger backbones and higher-resolution images (Luo et al., 2024; Hedlin et al., 2024; Li et al., 2023; Tang et al., 2023; Zhang et al., 2024).

In this paper, we propose MambaMatcher, a novel approach that overcomes the limitations of both feature-metric and correlation-metric methods by efficiently modeling high-dimensional correlation

maps using selective state-space models (SSMs). To the best of our knowledge, MambaMatcher is the first method to treat multi-level correlation scores at each position in the correlation map as a state in a state-space model, enabling effective and efficient modeling of inter-image correlations. At the core of MambaMatcher is a similarity-aware selective scan mechanism, which adapts Mamba's linear selective scanning algorithm to refine the 4D correlation tensor with linear complexity. This mechanism allows us to robustly and scalably process the correlation map without compromising on feature map resolution or receptive field, thereby capturing rich inter-image relationships while maintaining computational efficiency.

The key contributions of our work are summarized as follows:

- We introduce MambaMatcher, the first method to model high-dimensional correlation maps using selective state-space models, treating multi-level correlation scores as states to effectively capture inter-image correlations.
- We propose a novel similarity-aware selective scan mechanism, enabling efficient and accurate mining of inter-image correlations at high resolutions.
- MambaMatcher seamlessly integrates feature-metric and correlation-metric approaches into a unified pipeline, leveraging the strengths of both methods without compromising feature map resolution or receptive field.
- Extensive experiments demonstrate that MambaMatcher achieves state-of-the-art performance on standard semantic correspondence benchmarks, outperforming methods that rely on expensive Diffusion-based features, while incurring lower computational overhead.

## 2 RELATED WORK

**Feature-metric approach for semantic correspondence.** Semantic correspondence methods that adopt the feature-metric approach prioritize producing high-quality features to establish robust correspondences. Traditional feature-metric methods (Liu et al., 2010; Bristow et al., 2015; Cho et al., 2015; Ham et al., 2017) typically use hand-crafted descriptors (Lowe, 2004; Dalal & Triggs, 2005; Bay et al., 2006), which, despite their simplicity, show satisfactory performance. With the advent of deep learning, recent methods demonstrate that using local features extracted from deep neural networks leads to significant performance improvements (Min et al., 2020; Tang et al., 2023; Luo et al., 2024; Li et al., 2023). While ResNets (He et al., 2016) were the conventional choice for the visual feature extractor, more recent works propose employing stronger feature extractors such as DINOv2 (Oquab et al., 2023) or Stable Diffusion (Rombach et al., 2022). In the presence of supervision, there are attempts to yield richer features by refining the extracted features, *e.g.*, by using additional convolutional or attentional layers (Seo et al., 2018; Lee et al., 2021b; Huang et al., 2022). In our method, we leverage DINOv2 for its strong feature extraction capabilities and refine these features using 2D convolutional layers tailored to enhance correspondence accuracy. However, MambaMatcher takes a step further by harmoniously integrating the correlation-metric approach through our proposed similarity-aware selective scan, effectively modeling the correlation space and outperforming methods that use Stable Diffusion (Rombach et al., 2022) features.

**Correlation-metric approach for semantic correspondence.** Methods that adopt the correlation-metric approach aim to refine ambiguities and noise in the correlation map so that the refined map can be used to establish more robust and accurate correspondences. In the context of semantic correspondence, NCNet (Rocco et al., 2018) first established this idea via a 4D convolutional network to consider neighborhood consensus, which motivated follow-up work to formulate neighborhood consensus in more effective or efficient ways (Li et al., 2020; Min & Cho, 2021; Lee et al., 2021a; Kim et al., 2024). However, high-dimensional convolutional kernels are constrained by their local receptive field and static transformations. To address this, methods such as CATs (Cho et al., 2021; 2022) and TransforMatcher (Kim et al., 2022) apply the self-attention mechanism to the correlation tensor to consider inter-correlation relations in a dynamic global fashion. Despite their efficacy, applying self-attention to the correlation map incurs up to quartic computational complexity with respect to the feature map dimensions. This necessitates a compromise on either the feature map dimensions, the receptive field of inter-correlation relationship mining, or the expressivity of the algorithm used, leading to sub-optimal results compared to the recent success of feature-metric approaches.

In our work, we introduce a novel approach that models the correlation space using selective state-space models, applying a similarity-aware selective scan to the correlation tensor. Building upon the efficiency and scalability of Mamba (Gu & Dao, 2023), this method effectively overcomes previous limitations by avoiding compromises on feature map dimensions, receptive field, or network expressivity. Our approach facilitates a harmonious integration of feature-metric and correlation-metric techniques into a single pipeline, advancing the modeling of the correlation space.

**State-space models for computer vision.** State-space models use state variables to describe a system via a set of first-order differential or difference equations and were introduced into deep learning for sequence modeling (Gu et al., 2021b; Smith et al., 2022). The efficient leveraging of state-space models in deep learning gained rapid interest with the advent of Mamba (Gu & Dao, 2023), which showed promising results compared to transformer-based architectures for sequence modeling in natural language processing. Notably, Mamba exhibits linear computational complexity at inference, in contrast to attention-based methods that typically have quadratic complexity. This has inspired the application of the Mamba model to computer vision. VMamba (Liu et al., 2024b), Vision Mamba (Zhu et al., 2024), and PlainMamba (Yang et al., 2024) concurrently propose adopting the selective scan algorithm to the 2D image domain by varying the scan directions to accommodate spatial dimensions. These endeavors show competitive or superior performance compared to existing methods in the computer vision domain, motivating the application of Mamba to various downstream vision tasks such as video understanding (Li et al., 2024; Chen et al., 2024), medical imaging (Yue & Li, 2024; Ruan & Xiang, 2024), and point cloud understanding (Liu et al., 2024a). In our work, we extend the selective scan algorithm from Mamba by introducing a similarity-aware selective scan specifically designed to refine 4D correlation tensors. This adaptation enables us to effectively model the correlation space using selective state-space models, allowing for seamless handling of high-dimensional data in semantic correspondence tasks. By tailoring the selective scan to be similarity-aware, our method differs from previous applications by directly addressing the challenges of refining 4D correlation maps, which is critical for accurate semantic correspondence.

## 3 PRELIMINARY: SELECTIVE STATE SPACE MODELS (MAMBA)

State-space models (SSM) can be viewed as linear time-invariant (LTI) systems that maps a 1D function or sequence $x(t) \in \mathbb{R} \mapsto y(t)$ through a hidden state $h(t) \in \mathbb{R}^{\mathbf{N}}$. These models are mathematically formulated as linear ordinary differential equations (ODEs), with weighting parameters of $\mathbf{A} \in \mathbb{R}^{\mathbf{N} \times \mathbf{N}}, \mathbf{B} \in \mathbb{R}^{\mathbf{N} \times 1}, \mathbf{C} \in \mathbb{R}^{1 \times \mathbf{N}}$ and $D \in \mathbb{R}$:

$$
\begin{aligned}
h'(t) &= \mathbf{A}h(t) + \mathbf{B}x(t), \\
y(t) &= \mathbf{C}h(t) + Dx(t)
\end{aligned}
\tag{1}
$$

Recently, the key idea is to use the HiPPO matrix (Gu et al., 2020) for $\mathbf{A}$, which produces a hidden state that memorizes the sequence history. This is accomplished by tracking the coefficients of a Legendre polynomial, allowing the HiPPO matrix to approximate all of the previous history.

The S4 and Mamba are based on discrete versions of Eq.1, which include a timescale parameter $\Delta$ to transform the continuous parameters $\mathbf{A}, \mathbf{B}$ to discrete parameters $\overline{\mathbf{A}}, \overline{\mathbf{B}}$. The commonly used method for this transformation is the zero-order hold (ZOH), where the discretized result is:

$$
\begin{aligned}
\overline{\mathbf{A}} &= \exp(\Delta\mathbf{A}) \\
\overline{\mathbf{B}} &= (\Delta\mathbf{A})^{-1}(\exp(\Delta\mathbf{A}) - \mathbf{I}) \cdot \Delta\mathbf{B}
\end{aligned}
\tag{2}
$$

Consequently, the discretized version of Eq.1 using a step size of $\Delta$ can be rewritten as:

$$
\begin{aligned}
h_t &= \overline{\mathbf{A}}h_{t-1} + \overline{\mathbf{B}}x_t \\
y_t &= \mathbf{C}h_t + Dx_t
\end{aligned}
\tag{3}
$$

Structured State Space Model (S4) (Gu et al., 2021a) uses input-independent matrices $\mathbf{A}$, $\mathbf{B}$, and $\mathbf{C}$, allowing parallel computation via convolutional reformulation. However, this input-independence limits S4's efficacy compared to dynamic, input-dependent self-attention mechanisms. To overcome this, Mamba (Gu & Dao, 2023) introduces input dependency by making $\mathbf{B}$, $\mathbf{C}$, and the step size $\Delta$ functions of the input, allowing the model to dynamically adapt and enhancing effectiveness over static models. This content-awareness, termed *selective state-space models*, bridges the

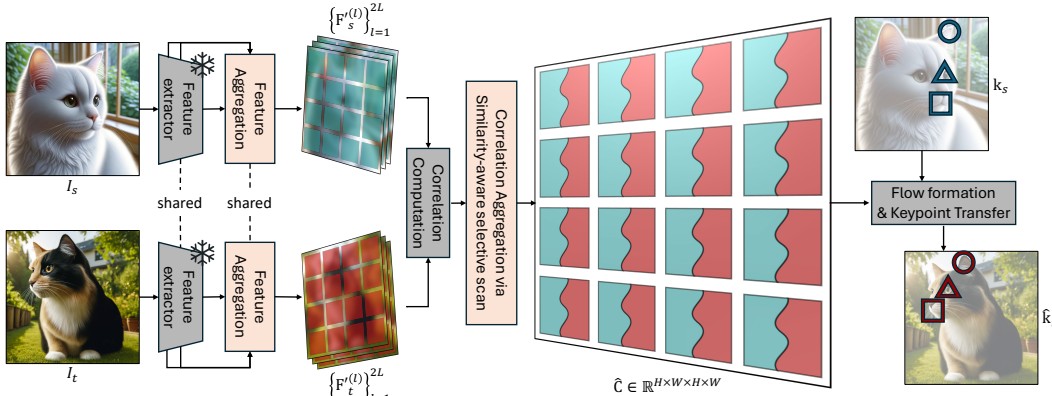

Figure 1: **Overview of MambaMatcher**. MambaMatcher extracts multi-level features for an image pair, which are processed using convolutional feature aggregation layers to yield improved features to compute the multi-level correlation map. The multi-level correlation map is processed using our proposed similarity-aware selective scan mechanism to yield a refined correlation map $\hat{\mathbf{C}}$, which can be used to establish semantic correspondences between the images.

gap between the efficiency of state-space models and the adaptability of self-attention. Although this precludes convolution representations with fixed kernels, Mamba achieves parallelization via a parallel scan algorithm based on associative scan algorithms (Martin & Cundy, 2017; Smith et al., 2022). This leads to the *selective scan algorithm*, which dynamically and efficiently scales linearly with sequence length, offering unbounded context and fast training and inference.

## 4 MAMBAMATCHER FOR SEMANTIC CORRESPONDENCE

We provide an overview of MambaMatcher in Fig. 1. Given a pair of images, we first extract multi-level feature maps from both images using a visual feature extractor. We then enhance these features using a simple yet effective convolutional feature aggregation module (Sec. 4.1). Next, we construct a multi-level correlation map from these features. This correlation map is refined using our correlation aggregation layers based on our novel similarity-aware selective scan (Sec. 4.2). Using the ground-truth source keypoints, we transfer them through the refined correlation map to obtain the *predicted* target keypoints (Sec. 4.3). Finally, we train the entire network by comparing these predicted keypoints with the *ground-truth* target keypoints (Sec. 4.4).

### 4.1 MULTI-LEVEL FEATURE EXTRACTION AND AGGREGATION

**Multi-level Feature Extraction.** Given a pair of images $(I_s, I_t)$, we utilize the pretrained DINOv2 ViT-B/14 (Oquab et al., 2023) as our visual feature extractor. We extract multi-level features from both the token and value representations across $L$ intermediate layers of the feature extractor, yielding $2L$ sets of features for each of the source and target images, i.e., $\{(\mathbf{F}_s^{(l)}, \mathbf{F}_t^{(l)})\}_{l=1}^{2L}$.

**Multi-level Feature Aggregation.** Prior to computing the correlation map, we enhance these multi-level features through feature aggregation to improve their self-awareness and robustness. This is achieved using a lightweight multi-layer 2D convolutional network specifically designed for this task. Formally, for each level $l$, the feature aggregation process is defined as:

$$\mathbf{F}'^{(l)} = \sigma\big(\mathbf{W}_2 * \big(\sigma(\mathbf{W}_1 * \mathbf{F}^{(l)})\big)\big) \tag{4}$$

where $\mathbf{W}_1$ and $\mathbf{W}_2$ are convolutional kernels, and $\sigma(\cdot)$ represents an activation function. As a result, we obtain $2L$ sets of aggregated features, $\{(\mathbf{F}_s'^{(l)}, \mathbf{F}_t'^{(l)})\}_{l=1}^{2L}$. The feature aggregators share the same weights across all levels to maintain efficiency.

Figure 2: **Correlation aggregation via similiarity-aware selective scan**. The multi-level correlation map $\mathbf{C}$ is flattened to form a sequence $\in \mathbb{R}^{(H \times W \times H \times W) \times 2L}$. This multi-level correlation sequence is sorted in the descending order of similarity scores, such that the selective scan can be performed in a *similarity-aware manner*. The refined correlation sequence is reordered to its original order, and is subsequently projected and reshaped to a single-level refined correlation map.

## 4.2 MULTI-LEVEL CORRELATION COMPUTATION AND AGGREGATION

**Multi-level Correlation Map Computation.** Using the refined features from the previous stage, we compute a correlation map $\mathbf{C}^{(l)} \in \mathbb{R}^{H \times W \times H \times W}$ for each level $l$:

$$\mathbf{C}^{(l)}(p_s, p_t) = \frac{\mathbf{F}_s'^{(l)}(p_s) \cdot \mathbf{F}_t'^{(l)}(p_t)}{\|\mathbf{F}_s'^{(l)}(p_s)\| \|\mathbf{F}_t'^{(l)}(p_t)\|} \tag{5}$$

where $p_s$ and $p_t$ denote spatial positions in the source and target feature maps, respectively, and $\|\cdot\|$ represents the L2 norm. The resulting $2L$ correlation maps are stacked to form a multi-level correlation map $\mathbf{C} \in \mathbb{R}^{2L \times H \times W \times H \times W}$.

**Correlation Aggregation via Similarity-aware Selective Scan.** Given the multi-level correlation map $\mathbf{C}$, we flatten it to form a *correlation sequence* $\overline{\mathbf{C}} \in \mathbb{R}^{(H \times W \times H \times W) \times 2L}$, enabling us to process it using the Mamba selective state-space model, which is adept at handling sequential data. Here, the $2L$ channels correspond to the similarity scores from each level; we treat these multi-level similarity scores as the 'state' in the selective SSM.

To embed similarity-awareness into the selective scan mechanism of Mamba, we propose scanning the correlation sequence in descending order of similarity scores. The rationale is that Mamba can retain relevant information over long sequences. By processing high-similarity regions first, we:

1. **Disambiguate High-Similarity Regions:** Early processing of strong matches helps resolve ambiguities in these regions.

2. **Refine Low-Similarity Regions:** Later stages can reinforce or diminish ambiguous or noisy correspondences by leveraging the context from earlier, more confident matches.

We sort the correlation sequence based on the similarity scores from the final correlation map (the $2L$-th level). After processing the sorted sequence with our similarity-aware selective scan mechanism, we reorder the sequence back to its original order. We then apply a linear projection and reshape to yield the refined correlation map $\hat{\mathbf{C}} \in \mathbb{R}^{H \times W \times H \times W}$.

## 4.3 KEYPOINT TRANSFER

To transfer keypoints from the source image to the target image, we transform the refined correlation tensor $\hat{\mathbf{C}}$ into a dense flow field using the kernel soft-argmax technique (Lee et al., 2019). Specifically, for each source keypoint position $(i, j)$, we apply a 2D Gaussian kernel $\mathbf{G}_{kl}^{\mathbf{P}}$ centered at $\mathbf{p} = \arg \max_{k,l} \hat{\mathbf{C}}_{ijkl}$ to promote a unimodal matching probability distribution, mitigating erroneous transfers due to ambiguous matches. We normalize the raw correlation outputs as follows:

$$\mathbf{C}^{\text{norm}}(i, j, k, l) = \frac{\exp\left(\mathbf{G}_{kl}^{\mathbf{P}} \hat{\mathbf{C}}(i, j, k, l)\right)}{\sum_{(k', l')} \exp\left(\mathbf{G}_{k'l'}^{\mathbf{P}} \hat{\mathbf{C}}(i, j, k', l')\right)} \tag{6}$$

Table 1: **Results of MambaMatcher on PF-PASCAL and SPair-71k datasets.** MambaMatcher outperforms existing baselines on both datasets, with reasonable latency and memory usage. We detail the backbone, supervision, and data augmentation usage of each method in Appendix B.

| Method | Image res. | PF-PASCAL $@\alpha_{\text{img}}$ | | | SPair-71k $@\alpha_{\text{bbox}}$ | | | time $(ms)$ | memory (GB) |
|---|---|---|---|---|---|---|---|---|---|
| | | 0.05 | 0.10 | 0.15 | 0.05 | 0.10 | 0.15 | | |
| DHPF (2020) | 240×240 | 75.7 | 90.7 | 95.0 | 20.9 | 37.3 | 47.5 | 58 | 1.6 |
| CHM (2021) | 240×240 | 80.1 | 91.6 | 94.9 | 27.2 | 46.3 | 57.5 | 54 | 1.6 |
| MMNet (2021) | 224×320 | 77.6 | 89.1 | 94.3 | - | 40.9 | - | 86 | - |
| PWarpC-NCNet (2022) | 400×400 | 79.2 | 92.1 | 95.6 | 31.6 | 52.0 | 61.8 | - | - |
| TransforMatcher (2022) | 240×240 | 80.8 | 91.8 | - | 32.4 | 53.7 | - | 54 | 1.6 |
| NeMF (2022) | 512×512 | 80.6 | 93.6 | - | 34.2 | 53.6 | - | 8500 | 6.3 |
| SCorrSAN (2022) | 256×256 | 81.5 | 93.3 | - | - | 55.3 | - | 28 | 1.5 |
| HCCNet (2024) | 240×240 | 80.2 | 92.4 | - | 35.8 | 54.8 | - | 30 | 2.0 |
| CATs++ (2022) | 512×512 | 84.9 | 93.8 | 96.8 | 40.7 | 59.8 | 68.5 | - | - |
| UFC (2023) | 512×512 | **88.0** | 94.8 | 97.9 | 48.5 | 64.4 | 72.1 | - | - |
| DIFT (2023) | 768×768 | 69.4 | 84.6 | 88.1 | 39.7 | 52.9 | - | - | - |
| DINO+SD$_{\text{zero-shot}}$ (2024) | 840² / 512² | 73.0 | 86.1 | 91.1 | - | 64.0 | - | - | - |
| DINO+SD$_{\text{sup}}$ (2024) | 840² / 512² | 80.9 | 93.6 | 96.9 | - | 74.6 | - | - | - |
| Diffusion Hyperfeatures (2024) | 224×224 | - | 86.7 | - | - | 64.6 | - | 6620 | - |
| Hedlin et al. (2024) | 0.93×ori. | - | - | - | 28.9 | 45.4 | - | 90k< | - |
| SD4Match (2023) | 768×768 | 84.4 | 95.2 | 97.5 | 59.5 | 75.5 | - | - | - |
| MambaMatcher (Ours) | 420×420 | 87.3 | **95.9** | **98.2** | **61.6** | **77.8** | **84.3** | 74 | 2.1 |

Using $\mathbf{C}^{\text{norm}}$, we transfer all coordinates on a dense grid $\mathbf{P} \in \mathbb{R}^{H \times W \times 2}$ corresponding to the source image $I_s$ to obtain their transferred coordinates $\hat{\mathbf{P}}$ on the target image $I_t$:

$$\hat{\mathbf{P}}(i,j) = \sum_{k,l} \mathbf{C}^{\text{norm}}(i,j,k,l) \cdot (k,l) \tag{7}$$

Here, $(k,l)$ represents spatial coordinates in the target image. $\mathbf{P}$ and $\hat{\mathbf{P}}$ are used to construct a dense flow field, which we employ to transfer source keypoints $\mathbf{k}_s$ to predicted target keypoints $\hat{\mathbf{k}}_t$.

### 4.4 TRAINING OBJECTIVE

Given an image pair $(I_s, I_t)$ with $M$ ground-truth keypoints, we use the above keypoint transfer scheme to obtain predicted target keypoints $\{\hat{\mathbf{k}}_t^{(m)}\}_{m=1}^M$. Our training objective is to minimize the average Euclidean distance between the predicted and ground-truth target keypoints:

$$\mathcal{L}_{\text{kp}} = \frac{1}{M} \sum_{m=1}^M \|\hat{\mathbf{k}}_t^{(m)} - \mathbf{k}_t^{(m)}\|_2^2 \tag{8}$$

Despite the simplicity of this loss function, our method achieves superior performance due to the effectiveness of the refined correlation map and the keypoint transfer process.

## 5 EXPERIMENTS

**Implementation details.** We use DINOv2 (ViT-B/14) (Oquab et al., 2023) as our visual feature extractor to obtain local features. We resize input images to $420 \times 420$, resulting in feature maps of size $H = W = 30$ and correlation maps of size $30^4$. Considering that ViT-B/14 has 12 transformer layers, we extract the token and value representations from layers 4 to 11, yielding a total of 8 layers $\times$ 2 facets = 16 feature maps for each image. These feature maps serve as inputs to our subsequent feature aggregation layer. Our feature aggregation layer consists of two layers of 2D convolution with a kernel size of 5, having output channel dimensions of 64 and 14, respectively, with a ReLU activation function in between. For the correlation aggregation layer, we build upon the open-source implementation of Mamba (Gu & Dao, 2023), using an SSM expansion factor of 16, local convolution width of 4, and block expansion factor of 3. We use the Adam optimizer (Kingma & Ba,

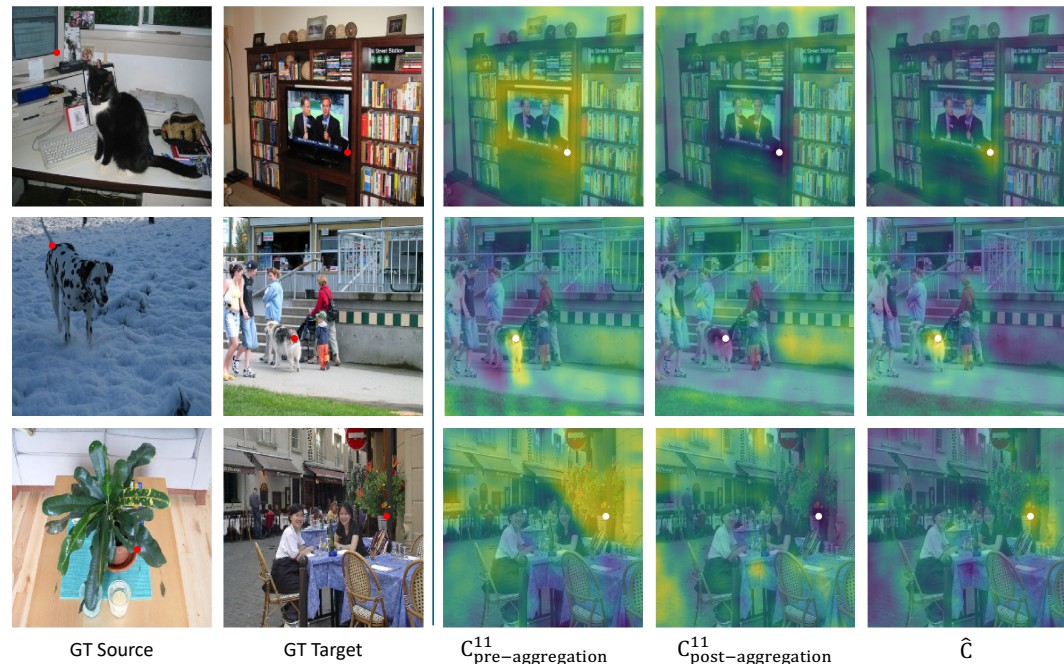

GT Source      GT Target      $C^{11}_{pre-aggregation}$      $C^{11}_{post-aggregation}$      $\hat{C}$

Figure 3: **Visualization of the effect of similarity-aware selective scan.** For each keypoint pair depicted in red on the left, we visualize the corresponding correlation map before and after the similarity-aware selective scan, and the final refined correlation. It shows that the refined correlation tensor can better localize (*i.e.,* has higher similarity, shown in brighter yellow) the keypoint position.

2014) with a constant learning rate of $1e-3$. We freeze the visual feature extractor during training to focus on learning the aggregation layers. MambaMatcher is implemented using PyTorch (Ansel et al., 2024) and PyTorch Lightning (Falcon & The PyTorch Lightning team, 2019).

**Evaluation metric.** We use the Percentage of Correct Keypoints (PCK), which is the standard evaluation metric for semantic correspondence. Given $M$ predicted and ground-truth target keypoint pairs $\mathcal{K} = \{(\hat{\mathbf{k}}_t^{(m)}, \mathbf{k}_t^{(m)})\}_{m=1}^M$, and a tolerance factor $\alpha_\tau$, PCK is measured by:

$$\text{PCK}(\mathcal{K}) = \frac{1}{M} \sum_{m=1}^M \mathbb{1}\left[\|\hat{\mathbf{k}}_t^{(m)} - \mathbf{k}_t^{(m)}\| \leq \alpha_\tau \cdot \max(w_\tau, h_\tau)\right], \quad (9)$$

where $w_\tau$ and $h_\tau$ are the width and height of either the image or the object bounding box.

### 5.1 PERFORMANCE ON SEMANTIC MATCHING

We evaluate MambaMatcher on the standard benchmarks for semantic matching: the PF-PASCAL (Ham et al., 2017) and SPair-71k (Min et al., 2019) datasets. The results are shown in Table 1, where MambaMatcher outperforms existing methods on both datasets, without relying on particularly large image sizes or computationally expensive backbones like Stable Diffusion. Moreover, our method incurs a reasonable computational overhead in terms of latency and memory usage.

Fig. 3 visualizes the effect of our proposed similarity-aware selective scan. We observe that our final refined correlation map $\hat{C}$ better localizes keypoints, as indicated by higher similarity scores (illustrated in brighter yellow). The initial correlation map prior to the similarity-aware selective scan shows high similarities at ground-truth locations, validating our choice of using the scores from the final correlation map to sort the multi-level correlation map. After aggregation via our similarity-aware selective scan, each level of the multi-level correlation map exhibits varying characteristics, which are condensed into our final refined correlation map $\hat{C}$ with enhanced keypoint localization. We provide additional details of this behavior in Appendix L.

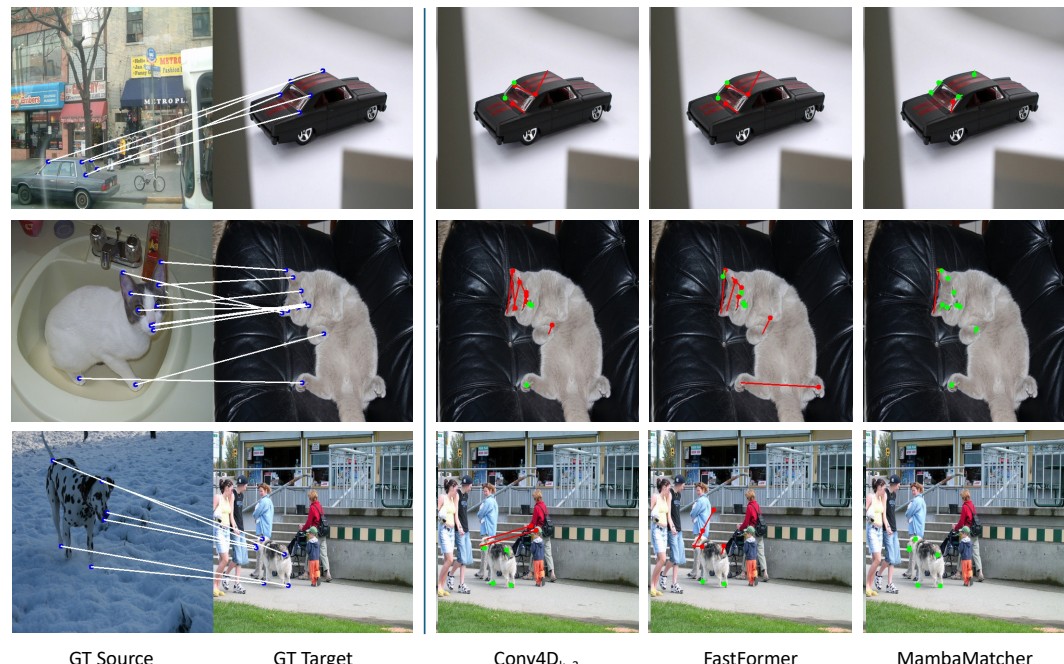

GT Source      GT Target      Conv4D$_{k=3}$      FastFormer      MambaMatcher

Figure 4: **Qualitative comparison to other correlation aggregation schemes**. Ground-truth correspondence are visualized in the left. The predicted keypoints are visualized on the right, where red depicts incorrect matches and green depicts correct matches. Our method shows to be more robust under large scale or viewpoint variations. Best viewed on electronics.

## 5.2 ANALYSIS ON THE FACET USED

Table 2: **Single facet comparison.**

| Facet | SPair-71k (s) $@\alpha_{bbox}$ | | |
|---|---|---|---|
| | 0.05 | 0.10 | 0.15 |
| Token | **25.2** | **43.1** | **55.7** |
| Query | 18.6 | 33.3 | 43.3 |
| Key | 15.6 | 30.0 | 41.4 |
| Value | 24.4 | 41.8 | 53.6 |

Table 3: **Facet combination comparison.**

| Facet used | SPair-71k (s) $@\alpha_{bbox}$ | | |
|---|---|---|---|
| | 0.05 | 0.10 | 0.15 |
| Token | 30.8 | 48.6 | 59.0 |
| + Value | 32.3 | 50.5 | 61.0 |
| + Value, Query | 32.5 | 50.6 | 61.0 |
| + Value, Query, Key | **33.2** | **50.7** | **61.3** |

When using DINOv2 as the feature extractor, we can utilize different facets: key, query, value, or token. In Table 2, we evaluate the PCK on the 'small' subset of SPair-71k when using each facet from the final layer (layer 11) of the DINOv2 backbone to establish a single-layer correlation map. We observe that the performance increases in the order of key, query, value, and token, demonstrating that the output token features are most effective for establishing semantic correspondences.

When using multi-level features (layers 4-11), we further experiment with incorporating additional features from other facets to improve performance. We default the multi-level correlation aggregation to a linear projection to a single-layer correlation map. The results in Table 3 show that using all feature sources results in the best performance. However, the most significant performance increase occurs when additionally using the value features; we observe a 1.9% increase at the 0.10 threshold compared to only a 0.1% increase when adding other facets. To balance performance and computational overhead, we choose to use token and value features, resulting in $2L$ layers of features when extracting features across $L$ layers.

Table 4: **Comparison between different feature aggregation schemes.**

| Feature aggregation | SPair-71k (s) @$\alpha_{\text{bbox}}$ | | |
|---|---|---|---|
| | 0.05 | 0.10 | 0.15 |
| None | 32.3 | 50.5 | 61.0 |
| 2D Conv$_{k=1}$ | 54.5 | 72.3 | 79.8 |
| 2D Conv$_{k=3}$ | 58.9 | 77.9 | 84.6 |
| 2D Conv$_{k=5}$ | **59.2** | **78.4** | **85.3** |
| Self-attn (2020) | 48.5 | 68.2 | 76.9 |
| + Cross-attn. | 35.2 | 57.0 | 68.6 |
| Mamba$_{\text{2D}}$ (2023) | 56.3 | 74.7 | 81.5 |
| + bidirectional | 53.3 | 75.2 | 82.8 |
| + Z-order curve (1966) | 56.0 | 75.2 | 82.8 |
| PlainMamba (2024) | 54.1 | 74.0 | 81.8 |

Table 5: **Comparison between different correlation aggregation schemes.**

| Correlation aggregation | SPair-71k (s) @$\alpha_{\text{bbox}}$ | | |
|---|---|---|---|
| | 0.05 | 0.10 | 0.15 |
| 4D Conv$_{k=1}$ | 59.2 | 78.4 | 85.3 |
| 4D Conv$_{k=3}$ | 59.2 | 78.2 | 85.2 |
| 4D Conv$_{k=5}$ | 39.2 | 67.9 | 79.0 |
| FastFormer (2022; 2021) | 59.5 | 78.9 | 85.7 |
| PlainMamba (2024) | 56.7 | 78.5 | 85.7 |
| Mamba$_{\text{4D}}$ | 59.3 | 78.8 | 85.6 |
| + bidirectional | 59.0 | 78.6 | 85.5 |
| + Z-order curve (1966) | 58.4 | 79.0 | 85.6 |
| + ascending order | 58.4 | 78.2 | 85.2 |
| + descending order | **59.9** | **79.3** | **86.2** |

## 5.3 ANALYSIS ON FEATURE AND CORRELATION AGGREGATION

**Feature aggregation analysis.** Table 4 presents the comparative performance when using different feature aggregation schemes, evaluated on the *small* subset of SPair-71k. As we extract multi-level features and compute multi-level correlation maps, we obtain a single-level refined correlation map after feature aggregation using a single $1 \times 1$ convolution layer in these experiments. Using no feature aggregation ('None') defaults to using the extracted features directly. When applying Mamba$_{\text{2D}}$, we flatten the multi-level feature map $\mathbf{F} \in \mathbb{R}^{2L \times H \times W}$ to a sequence in $\mathbb{R}^{2L \times (H \times W)}$, which is then input to a Mamba layer. We also experiment with bi-directional selective scans, considering that an image does not have a fixed beginning or end, unlike a temporal sequence. The Z-order curve (Morton, 1966) is a representative space-filling curve that forms a path passing through every point in a high-dimensional discrete space while preserving spatial proximity, which has been effective in prior work (Wu et al., 2023; Liang et al., 2024) and is applicable to the scan order of Mamba. Among convolution-based, attention-based, and Mamba-based feature aggregation schemes, we find that using a series of 2D convolutional layers with a kernel size of 5 performs the best.

**Correlation aggregation analysis.** Table 5 illustrates the comparative performance when using different correlation aggregation schemes, evaluated on the *small* subset of SPair-71k. Based on the results of the feature aggregation comparison (Table 4), we default the feature aggregation scheme to 2D Conv$_{k=5}$ for these experiments. Applying a vanilla transformer (Dosovitskiy et al., 2020) to the correlation tensor results in out-of-memory errors even on a single batch on an RTX 3090 GPU; therefore, we opt for FastFormer (Wu et al., 2021) as the linear-complexity attention-based correlation aggregation scheme (Kim et al., 2022). When applying Mamba$_{\text{4D}}$, we flatten the multi-level correlation map $\mathbf{C} \in \mathbb{R}^{L \times H \times W \times H \times W}$ to a sequence in $\mathbb{R}^{L \times (H \times W \times H \times W)}$, which is then input to a Mamba layer. 'Ascending order' and 'Descending order' indicate that the flattened correlation tensor is sorted in either ascending or descending order based on the similarity scores from the final correlation map of the multi-level correlation map. Among different schemes, we find that processing a descending-order sorted multi-level correlation map shows to be the best alternative to induce similarity-awareness in the selective scan algorithm, verifying the design choices of MambaMatcher. Fig. 4 compares different correlation aggregation schemes, showing that MambaMatcher establishes more robust and accurate semantic correspondences under large viewpoint or scale variations. We provide qualitative comparisons between different selective scanning schemes in Fig. 5. We include further analyses and comparisons in Appendix C, D, E and J.

## 5.4 ANALYSIS ON EFFICIENCY OF MAMBAMATCHER

For an intuitive overview, we measure module-wise maximum GPU memory usage and latency in Table 6. The values are cumulative in the order of DINOv2 (feature extraction), feature aggrega-

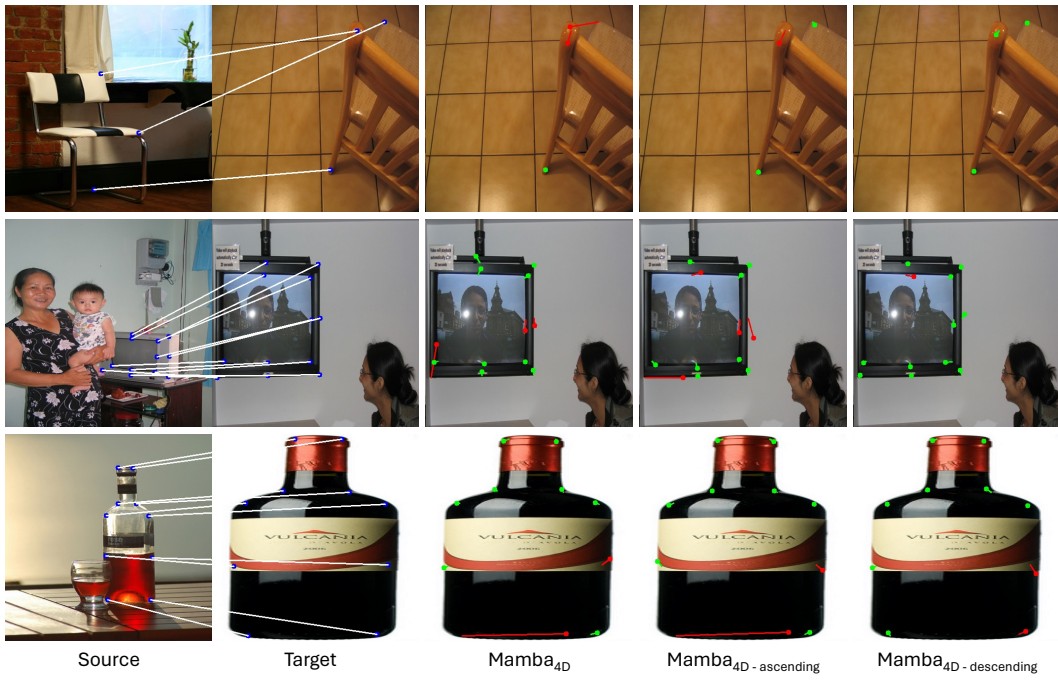

Figure 5: **Comparison of Mamba$_{4D}$ scanning schemes.** It can be seen that our choice of scanning the correlation sequence in a descending order shows better keypoint localization, evidencing improved denoising and disambiguation of the correlation sequence.

tion, and correlation aggregation. This shows that our design incurs the lowest latency while using less memory and fewer parameters than FastFormer, demonstrating a favorable balance between computational overhead and performance[1].

Table 6: **Memory, Latency and # Params comparison across correlation schemes.** Our scheme strikes the most favorable balance between performance and efficiency.

| Module | GPU Memory (GB) | Latency (ms) | # Params |
|---|---|---|---|
| DINOv2 (Oquab et al., 2023) | 0.97 | 10.3 | 86.6M |
| Feature aggregation | 1.17 | 12.3 | 42.5M |
| Correlation aggregation | | | |
| - Conv4D$_{k=3}$ | 1.17 | 41.0 | 1.3K |
| - FastFormers (Kim et al., 2022) (6 layers) | 1.67 | 28.8 | 26.0K |
| - Mamba$_{4D}$ + Similarity-aware Selective Scan (Ours) | 1.64 | 16.4 | 5.1K |

## 6 CONCLUSION

In conclusion, we introduced MambaMatcher, a novel approach for semantic correspondence that models the high-dimensional correlation space using selective state-space models (SSMs), treating multi-level correlation scores as states within the correlation map. By leveraging the efficiency of Mamba's linear-complexity algorithm and implementing a similarity-aware selective scan mechanism, MambaMatcher effectively refines 4D correlation tensors without compromising feature map resolution or receptive field. Our evaluations on standard benchmarks demonstrate that MambaMatcher significantly enhances keypoint localization by inducing high similarity values near true keypoint positions, outperforming existing methods while maintaining computational efficiency. This work not only advances the state of the art in semantic correspondence but also highlights the potential of applying SSMs to high-dimensional data, encouraging further exploration into integrating feature-metric and correlation-metric approaches in visual correspondence tasks.

---

[1]We report the FLOPs of MambaMatcher in Appendix E.

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

## A  ADDITIONAL IMPLEMENTATION DETAILS

During training of MambaMatcher in Tab. 1, we use an effective batch size of 80 by distributing 10 batches to 8 RTX 3090 GPUs. For other comparison and ablative experiments, we run the experiments on a 'small' subset of SPair-71k, which is around 20% the size of the original SPair-71k dataset (Min et al., 2019), with varying effective sizes across 2 GPUs. The batch sizes vary because different feature and correlation aggregation schemes required different amount of vRAM. For example, when using FastFormer (Wu et al., 2021), only 3 batches could fit into a single GPU when training.

**Details of soft sampler (Sec. 4.3.** Given a source keypoint $\mathbf{k}_s = (x_{k_s}, y_{k_s})$, we define a soft sampler $\mathbf{W}^{\mathbf{k}_s} \in \mathbb{R}^{H \times W}$:

$$\mathbf{W}^{\mathbf{k}_s}(i,j) = \frac{\max(0, \tau - \sqrt{(x_{k_s} - j)^2 + (y_{k_s} - i)^2})}{\sum_{i'j'} \max(0, \tau - \sqrt{(x_{k_s} - j')^2 + (y_{k_s} - i')^2})} \tag{10}$$

where $\tau$ is a distance threshold from the keypoint, and $\sum_{ij} \mathbf{W}^{k_s}(i,j) = 1$. The role of the soft sampler is to sample each transferred keypoint $\hat{\mathbf{P}}(i,j)$ by assigning weights which are inversely proportional to the distance to the keypoint $\mathbf{k}_s$. We can obtain sub-pixel accurate keypoint matches as follows:

$$\hat{\mathbf{k}}_t = \sum_{(i,j) \in H \times W} \hat{\mathbf{P}}(i,j) \mathbf{W}^{k_s}(i,j). \tag{11}$$

We use $\tau = 0.1$ for training, and $\tau = 0.05$ for inference.

**Experimental environment.** All experiments are run on a machine with an Intel(R) Xeon(R) Gold 6242 CPU, with up to 8 GeForce RTX 3090 GPUs.

## B  ADDITIONAL DETAILS OF BASELINE METHODS

We provide the details of each baseline approach (shown in Table 1 of the main manuscript) in Table 7, which was omitted due to spatial constraints.

Table 7: **Additional details of baseline methods.**

| Method | Feature backbone | Supervision | Data augmentation |
|---|---|---|---|
| DHPF, CHM, MMNet, PWarpC-NCNet, NeMF, SCorrSAN | ResNet101 | kp-pair | x |
| TransforMatcher, CATs++, HCCNet, UFC | ResNet101 | kp-pair | o |
| DIFT | SD2.1 | None | x |
| DINO + SD$_{\text{zero-shot}}$ | DINOv2, SD1.5 | None | x |
| DINO + SD$_{\text{supervised}}$ | DINOv2, SD1.5 | kp-pair | x |
| Diffusion Hyperfeatures | SD1.5 | None | x |
| Hedlin et al. (2024) | SD1.4 | None | x |
| SD4Match | SD2.1 | kp-pair | x |
| MambaMatcher | DINOv2 | kp-pair | o |

## C  FEATURE BACKBONE / DATA AUGMENTATION COMPARISON

Table 8: **PCK of MambaMatcher on SPair-71k when using varying feature backbones and data augmentation.** We follow the data augmentation scheme used in CATs (Cho et al., 2021) and TransforMatcher (Kim et al., 2022)

| Backbone | Data aug. | PCK@0.05 | PCK@0.10 | PCK@0.15 |
|---|---|---|---|---|
| ResNet101 | x | 38.2 | 53.3 | 61.3 |
| ResNet101 | o | 41.0 | 58.5 | 67.4 |
| DINOv2 | x | 57.9 | 74.6 | 81.8 |
| DINOv2 | o | 61.6 | 77.8 | 84.3 |

We provide the results of MambaMatcher when using varying backbones, with or without data augmentation, on SPair-71k for a fairer comparison in Table 8. Noting that PCK@0.05/0.10 for TransforMatcher (Kim et al., 2022) are 32.4/53.7 with data augmentation, these results show that the similarity-aware selective scan shows enhanced efficacy over multiple layers of additive attention (FastFormers (Wu et al., 2021)).

# D  STATISTICAL SIGNIFICANCE OF PERFORMANCE GAP IN COMPARISON TO FASTFORMERS

We conduct 3 repeated experiments with varying seeds to report the mean and variance of PCK results on the 'small' subset of SPair-71k in Table 9. While the performance gain is not dramatic, MambaMatcher offers advantages in terms of computational overhead (memory, latency) as previously shown in Table 6.

Table 9: **PCK results on SPair-71K over multiple runs** We report the results when using Fast-Formers in comparison to our similarity-aware selective scan as the correlation aggregation. The experiments were conducted 3 times - the mean and standard variation across the runs are reported. It can be seen that our scheme consistently yields better performances across PCK thresholds.

| Method | PCK@0.05 | PCK@0.10 | PCK@0.15 |
|---|---|---|---|
| FastFormers (Kim et al., 2022) (6 layers) | $59.9 \pm 0.74$ | $76.9 \pm 1.40$ | $83.9 \pm 1.27$ |
| Mamba + Similarity-aware Selective Scan (Ours) | $60.6 \pm 0.54$ | $78.2 \pm 0.76$ | $85.0 \pm 0.86$ |

# E  FLOPS ANALYSIS OF MAMBAMATCHER

In the Table 10, we report the FLOPs of MambaMatcher using open-source libraries `ptflops` and `calflops`.

Table 10: **FLOPs of MambaMatcher measured using open-source libraries.**

| Module | ptflops | calflops |
|---|---|---|
| DINOv2 (Oquab et al., 2023) | 359.32G | 358.99G |
| Feat. agg | 2.45T | 2.45 T |
| Conv4D$_{k=3}$ (Min & Cho, 2021) | 2.06G | 2.06G |
| FastFormers (Kim et al., 2022) (6 layers) | 43.54G | 43.05G |
| Mamba + Similarity-aware Selective Scan (Ours) | 27.54M | 3.84G |

While FLOPs serve as a standardized measure of computational complexity, we noticed that existing libraries fail to accurately capture the FLOPs of various modules due to technical complexities, *e.g.,* reliance on operations registered as `nn.Modules`. Additionally, certain libraries for measuring FLOPs crash when encountered with hardware-optimized algorithms from `xFormers` (Lefaudeux et al., 2022), which are used in the DINOv2 backbone of our method. Consequently, we believe that this measurement may not be entirely fair or representative of the actual computational overhead and efficiency.

To address this gap, we conduct a theoretical calculation of FLOPs for varying correlation aggregation schemes. We consider an input with dimensions $N \times C = 30^4 \times 16$, consistent with MambaMatcher. We assume the same dimensions for the input and output *i.e.,* $C = C_{\text{in}} = C_{\text{out}}$.

**4D convolution, kernel size 3**.
$2 \times N \times C_{\text{in}} \times C_{\text{out}} \times k^4 = 33.6$ GFLOPs

**Vanilla dot-product attention**. Assuming single head, QKV dim = 16.
QKV projection: $3 \times (2 \times N \times C_{\text{in}} \times C_{\text{out}})$
Dot-product: $2 \times (N^2 \times C)$
Softmax: $3 \times (N^2)$

Weighted sum of V: $2 \times (N^2) \times C$
Total = 44.0 TFLOPs

**FastFormers (Additive attention)**. Assuming single head, QKV dim = 16.
QKV projection: $3 \times (2 \times N \times C_{\text{in}} \times C_{\text{out}})$
Softmax and weighted sum: $2 \times (3 \times N + 2 \times N \times C)$
Global vector addition: $2 \times (N \times C)$
Projection: $2 \times N \times C^2$ Total = 1.74 GFLOPs

**Mamba: selective state-space machines**. Hyperparameters following MambaMatcher.
Input projection: $2 \times 2 \times N \times C_{\text{in}} \times C_{\text{inner}}$
1D convolution: $2 \times C_{\text{inner}} \times k \times C_{\text{inner}}$
Projection to A, B, dt: $2 \times N \times C_{\text{inner}} times (2 \times d_{\text{model}} + 1)$
Selective scan: $9 \times N \times d_{\text{model}} \times d_{\text{state}}$
Element-wise multiplication: $N \times C_{\text{inner}}$
Output projection: $2 \times N \times C_{\text{inner}} \; times C_{\text{in}}$
Total FLOPs = 23.1 GFLOPs

**Ours: Selective state-space machines with Similarity-aware Selective Scan**. Same as above, but additional sorting overhead. Assuming each comparison and swap operation involves approximately 4 FLOPs:
Sorting: $4 \times (N log N) = 0.064$GFLOPs
Total FLOPs = 23.2 GFLOPs

Note that the above values ignore many details, including activation, normalization, residual connections, or actual number of aggregation layers used. The above theoretical calculation serve to provide a vague estimate of FLOPs for each scheme. However, we suggest that the number of FLOPs does not directly translate to computational overhead in learning-based methods, as many variables such as parallelism, hardware optimization, and intermediate representations directly impact GPU memory usage and latency.

# F    GENERALIZABILITY OF MAMBAMATCHER

**Trained on PF-PASCAL, evaluated on PF-WILLOW.** We present the results of MambaMatcher on the PF-WILLOW (Ham et al., 2017) dataset. The PF-WILLOW dataset contains 900 image pairs for testing only and is evaluated using the model trained on the PF-PASCAL dataset. The results are illustrated in Table 11, where it can be seen that while MambaMatcher performs competitively, it does not outperform existing methods. This is unlike our results on the PF-PASCAL and SPair-71k datasets (Table 1), where MambaMatcher outperforms all existing benchmarks. This may be attributed to supervised training, which causes the feature and correlation aggregation layers to be trained specifically for the training domain. Another possibility is that the Mamba layer lacks generalizability to unseen domains compared to other methods built on convolutional or attention-based layers.

**Trained on SPair-71k, evaluated on PF-PASCAL.** While we provide the generalization performance of MambaMatcher on the PF-WILLOW dataset in Table 11, we report additional generalization results in Table 12. Results on PF-PASCAL were trained on SPair-71k, and vice versa. The results indicate that while the generalizability of MambaMatcher is not state-of-the-art, it generalizes competitively with other state-of-the-art methods in certain cases, such as being trained on PF-PASCAL and tested on SPair-71k. While domain generalization is advantageous, we suggest that a lack of cross-dataset generalization does not diminish the overall significance of our method. If large-scale datasets for semantic correspondence become available, this problem is likely to be alleviated significantly for all semantic matching methods.

# G    COMPARISON ON THE DINOv2 LAYERS USED

We show the comparative experiments on the layers if DINOv2 used in this work to validate our use of layers 4-11. The experiments were carried out on the 'small' set of SPair-71k. The results

Table 11: **Results of MambaMatcher on the PF-WILLOW dataset.** We perform competitively with existing methods, but do not outperform all existing methods unlike on PF-PASCAL or SPair-71k.

| Method | PF-WILLOW | | | |
| | @$\alpha_{\text{bbox}}$ | | @$\alpha_{\text{bbox-kp}}$ | |
| | 0.05 | 0.10 | 0.05 | 0.10 |
| --- | --- | --- | --- | --- |
| DHPF (2020) | 49.5 | 77.6 | - | 71.0 |
| CHM (2021) | 52.7 | 79.4 | - | 69.6 |
| CATs++ (2022) | 56.7 | 81.2 | 47.0 | 72.6 |
| PWarpC-NCNet (2022) | - | - | 48.0 | 76.2 |
| TransforMatcher (2022) | - | 76.0 | - | 65.3 |
| NeMF (2022) | - | - | 60.8 | 75.0 |
| SCorrSAN (2022) | 54.1 | 80.0 | - | - |
| HCCNet (2024) | - | 74.5 | - | 65.5 |
| UFC (2023) | 58.6 | 81.2 | 50.4 | 74.2 |
| DIFT (2023) | 58.1 | 81.2 | 44.8 | 68.0 |
| DINO+SD$_{\text{zero-shot}}$ (2024) | - | - | - | - |
| DINO+SD$_{\text{sup}}$ (2024) | - | - | - | - |
| Diffusion Hyperfeatures (2024) | - | 78.0 | - | - |
| Hedlin et al. (2024) | 53.0 | 84.3 | - | - |
| SD4Match (2023) | - | - | 52.1 | 80.4 |
| Ours | 56.2 | 81.1 | 47.4 | 72.1 |

Table 12: **PCK on SPair-71k after being trained on PF-PASCAL.**

| Model | PCK@0.05 | PCK@0.10 | PCK@0.15 |
| --- | --- | --- | --- |
| CATs (Cho et al., 2021) | 13.6 | 27.0 | - |
| TransforMatcher (Kim et al., 2022) | - | 30.1 | - |
| SD4Match (Li et al., 2023) | 27.2 | 40.9 | - |
| MambaMatcher (Ours) | 26.5 | 40.9 | 49.1 |

in Tab. 13 shows that better features can be obtained across the depths of the DINOv2 backbone, with the 11th layer token features exhibiting the best performance. Tab. 14 aims to choose the best combination of layers to extract the feature maps from. While the PCK performance improves gracefully as more layers are used, we choose to use layers 4-11 as the performance improvement beyond that becomes diminishing, and using layers 4-11 provides us with a favorable compromise between memory usage (around 70% memory usage compared to using all 0-11 layers) and PCK performance.

## H  PCK PER IMAGE V.S. PCK PER POINT

While it is conventional to calculate the mean PCK per image (sum of image-wise PCK averaged over the number of images) when reporting the PCK results, some methods confuse this concept with PCK per point (sum of pair-wise PCK averaged over the number of point pairs). Tab. 15 shows the results, where it can be seen that PCK-per-point yields higher values in comparison.

## I  PCK PER CATEGORY

We present the category-wise PCK in Tab. 16, where it can be seen that MambaMatcher yields the best results overall.

## J  POTENTIAL WHEN USING LARGER RESOLUTIONS

In Table 17, we report the GPU memory / latency usage when using different correlation aggregation module at varying image resolutions (thus, varying feature and correlation map resolutions).

Table 13: **Comparison between different layers of the DINOv2 backbone.**

| | SPair-71k (s) | | |
|---|---|---|---|
| Layers used | $@\alpha_{img}$ | | |
| | 0.05 | 0.10 | 0.15 |
| 0 | 0.9 | 3.8 | 8.2 |
| 1 | 1.5 | 5.3 | 11.2 |
| 2 | 1.7 | 6.1 | 12.2 |
| 3 | 4.2 | 11.1 | 18.8 |
| 4 | 7.3 | 16.1 | 24.6 |
| 5 | 10.2 | 20.6 | 29.8 |
| 6 | 13.1 | 23.6 | 31.8 |
| 7 | 17.5 | 29.8 | 39.0 |
| 8 | 20.7 | 35.2 | 45.7 |
| 9 | 23.9 | 40.3 | 51.5 |
| 10 | 25.2 | 42.5 | 54.1 |
| 11 | 25.2 | 43.1 | 55.7 |

Table 14: **Comparison between different layers combinations of the DINOv2 backbone.**

| | SPair-71k (s) | | |
|---|---|---|---|
| Layers used | $@\alpha_{img}$ | | |
| | 0.05 | 0.10 | 0.15 |
| 11 | 25.2 | 43.1 | 55.7 |
| 10-11 | 29.2 | 46.4 | 56.8 |
| 9-11 | 28.9 | 46.7 | 58.0 |
| 8-11 | 29.6 | 47.4 | 58.3 |
| 7-11 | 30.4 | 48.5 | 58.8 |
| 6-11 | 30.8 | 48.4 | 58.7 |
| 5-11 | 30.9 | 48.4 | 58.6 |
| 4-11 | 30.8 | 48.6 | 59.0 |
| 3-11 | 31.0 | 48.7 | 59.0 |
| 2-11 | 31.2 | 48.9 | 58.7 |
| 1-11 | 31.4 | 48.9 | 58.8 |
| 0-11 | 31.4 | 48.9 | 58.7 |

Table 15: **Results of MambaMatcher on PF-PASCAL and SPair-71k datasets.** MambaMatcher outperforms existing baselines on both datasets. MambaMatcher * outperforms MambaMatcher, showing that PCK-per-point yields higher results in comparison to PCK-per-image.

| Method | Image res. | PF-PASCAL $@\alpha_{img}$ | | | SPair-71k $@\alpha_{bbox}$ | | | time (*ms*) | memory (GB) |
|---|---|---|---|---|---|---|---|---|---|
| | | 0.05 | 0.10 | 0.15 | 0.05 | 0.10 | 0.15 | | |
| DHPF (2020) | 240×240 | 75.7 | 90.7 | 95.0 | 20.9 | 37.3 | 47.5 | 58 | 1.6 |
| CHM (2021) | 240×240 | 80.1 | 91.6 | 94.9 | 27.2 | 46.3 | 57.5 | 54 | 1.6 |
| MMNet (2021) | 224×320 | 77.6 | 89.1 | 94.3 | - | 40.9 | - | 86 | - |
| PWarpC-NCNet (2022) | 400×400 | 79.2 | 92.1 | 95.6 | 31.6 | 52.0 | 61.8 | - | - |
| TransforMatcher (2022) | 240×240 | 80.8 | 91.8 | - | 32.4 | 53.7 | - | 54 | 1.6 |
| NeMF (2022) | 512×512 | 80.6 | 93.6 | - | 34.2 | 53.6 | - | 8500 | 6.3 |
| SCorrSAN (2022) | 256×256 | 81.5 | 93.3 | - | - | 55.3 | - | 28 | 1.5 |
| HCCNet (2024) | 240×240 | 80.2 | 92.4 | - | 35.8 | 54.8 | - | 30 | 2.0 |
| CATs++ (2022) | 512×512 | 84.9 | 93.8 | 96.8 | 40.7 | 59.8 | 68.5 | - | - |
| UFC (2023) | 512×512 | **88.0** | 94.8 | 97.9 | 48.5 | 64.4 | 72.1 | - | - |
| DIFT (2023) | 768×768 | 69.4 | 84.6 | 88.1 | 39.7 | 52.9 | - | - | - |
| DINO+SD$_{zero-shot}$ (2024) | 840² / 512² | 73.0 | 86.1 | 91.1 | - | 64.0 | - | - | - |
| DINO+SD$_{sup}$ (2024) | 840² / 512² | 80.9 | 93.6 | 96.9 | - | 74.6 | - | - | - |
| Diffusion Hyperfeatures (2024) | 224×224 | - | 86.7 | - | - | 64.6 | - | 6620 | - |
| Hedlin et al. (2024) | 0.93×ori. | - | - | - | 28.9 | 45.4 | - | 90k< | - |
| SD4Match (2023) | 768×768 | 84.4 | 95.2 | 97.5 | 59.5 | 75.5 | - | - | - |
| MambaMatcher (Ours) | 420×420 | 87.3 | 95.9 | **98.2** | 61.6 | 77.8 | 84.3 | 74 | 2.1 |
| MambaMatcher * (Ours) | 420×420 | 87.6 | **96.0** | **98.2** | **63.3** | **79.2** | **85.6** | 74 | 2.1 |

We do not report the PCK results, because the images were simply resized and the networks were not trained on those image sizes. Note that the memory usage is cumulative i.e., maximum GPU memory usage during the forward run. It can be seen that our similarity-aware selective scan incurs

Table 16: **Category-wise PCK on the SPair-71k dataset.**

| Method | Aero | Bike | Bird | Boat | Bottle | Bus | Car | Cat | Chair | Cow | Dog | Horse | Motor | Person | Plant | Sheep | Train | TV | All |
|---|---|---|---|---|---|---|---|---|---|---|---|---|---|---|---|---|---|---|---|
| DINOv2 (2023) | 69.9 | 58.9 | 86.8 | 36.9 | 43.4 | 42.6 | 39.3 | 70.2 | 37.5 | 69.0 | 63.7 | 68.9 | 55.1 | 65.0 | 33.3 | 57.8 | 51.2 | 31.2 | 53.9 |
| DIFT (2023) | 61.2 | 53.2 | 79.5 | 31.2 | 45.3 | 39.8 | 33.3 | 77.8 | 34.7 | 70.1 | 51.5 | 57.2 | 50.6 | 41.4 | 51.9 | 46.0 | 67.6 | 59.5 | 52.9 |
| SD+DINO (2024) | 71.4 | 59.1 | 87.3 | 38.1 | 51.3 | 43.3 | 40.2 | 77.2 | 42.3 | 75.4 | 63.2 | 68.8 | 56.0 | 66.1 | 52.8 | 63.0 | 55.1 | 59.3 | |
| NCNet (2018) | 17.9 | 12.2 | 32.1 | 11.7 | 29.0 | 19.9 | 16.1 | 39.2 | 9.9 | 23.9 | 18.8 | 15.7 | 17.4 | 15.9 | 14.8 | 9.6 | 24.2 | 31.1 | 20.1 |
| PMNC (2021a) | 54.1 | 35.9 | 74.9 | 36.5 | 42.1 | 48.8 | 40.0 | 72.6 | 21.1 | 67.6 | 58.1 | 50.5 | 40.1 | 54.1 | 43.3 | 35.7 | 74.5 | 59.9 | 50.4 |
| TransforMatcher (2022) | 59.2 | 39.3 | 73.0 | 41.2 | 52.5 | 66.3 | 55.4 | 67.1 | 26.1 | 67.1 | 56.6 | 53.2 | 45.0 | 39.9 | 42.1 | 35.3 | 75.2 | 68.6 | 53.7 |
| SCorrSAN (2022) | 57.1 | 40.3 | 78.3 | 38.1 | 51.8 | 57.8 | 47.1 | 67.9 | 25.2 | 71.3 | 63.9 | 49.3 | 45.3 | 49.8 | 48.8 | 40.3 | 77.7 | 69.7 | 55.3 |
| SD4Match (2023) | 75.3 | 67.4 | 85.7 | 64.7 | 62.9 | 86.6 | 76.5 | 82.6 | 64.8 | 86.7 | 73.0 | 78.9 | 70.9 | 78.3 | 66.8 | 64.8 | 91.5 | 86.6 | 75.5 |
| MambaMatcher (Ours) | 82.9 | 61.0 | 91.9 | 61.0 | 62.7 | 89.9 | 83.8 | 89.9 | 60.6 | 86.7 | 81.2 | 81.6 | 73.7 | 79.5 | 70.0 | 71.5 | 93.0 | 86.4 | **77.8** |

Large viewpoint variation + symmetry ambiguity          Multiple instances + semantic ambiguity

Figure 6: **Failure case of MambaMatcher.** We analyze the common failure cases of our method. Firstly, MambaMatcher shows to fail more often in the dual presence of large viewpoint variation and symmetry ambiguity, where our model fails to accurately distinguish the position given a symmetric instance. Secondly, MambaMatcher often fails to follow the ground-truth in the dual presence of multiple instances and semantic ambiguity. For example, in the upper-right image, an eye and the nose of the sheep is predicted to correspond to an eye and the nose of a neighbouring dog.

consistently lower GPU memory usage and latency compared to FastFormers. Most notably, the difference in latency is dramatic; the hardware optimizations of Mamba enables the similarity-aware selective scan to be performed with only a small increase in latency even when the image sizes become significantly larger. This further justifies our usage of Mamba, given larger image inputs i.e., consequently, longer correlation sequences.

Table 17: **Efficiency comparison when using larger image resolutions.**

| Image res. | Feature res. | Correlation agg. | GPU memory (GB) | latency (ms) |
|---|---|---|---|---|
| 420 | 30 | Ours | 1.64 | 16.4 |
| 420 | 30 | FastFormer | 1.67 | 28.8 |
| 560 | 40 | Ours | 3.25 | 16.6 |
| 560 | 40 | FastFormer | 3.16 | 28.9 |
| 700 | 50 | Ours | 6.47 | 17.7 |
| 700 | 50 | FastFormer | 6.27 | 55.6 |

### J.1 FAILURE CASE ANALYSIS

We include qualitative examples of failure cases of our method in Figure 6. Firstly, MambaMatcher tends to fail in scenarios involving large viewpoint variations combined with symmetry ambiguity, where our model struggles to accurately distinguish positions in symmetric instances. Secondly, MambaMatcher may not follow the ground truth in the presence of multiple instances and semantic ambiguity. We suggest that incorrect correspondence predictions due to semantic ambiguity could still be considered as semantic correspondences in a broader sense. This opens up interesting future directions, such as exploring many-to-many semantic correspondences instead of just one-to-one correspondences in existing datasets.

## K  ADDITIONAL QUALITATIVE RESULTS

We provide additional qualitative results in Fig. 7.

## L  ADDITIONAL VISUALIZATIONS OF REFINED CORRELATION MAP

We provide additional visualizations of refined correlations in Fig. 8. While Fig. 3 demonstrates that our refined correlation map can better localize keypoints, it also shows that the $\mathbf{C}^{11}$, post-

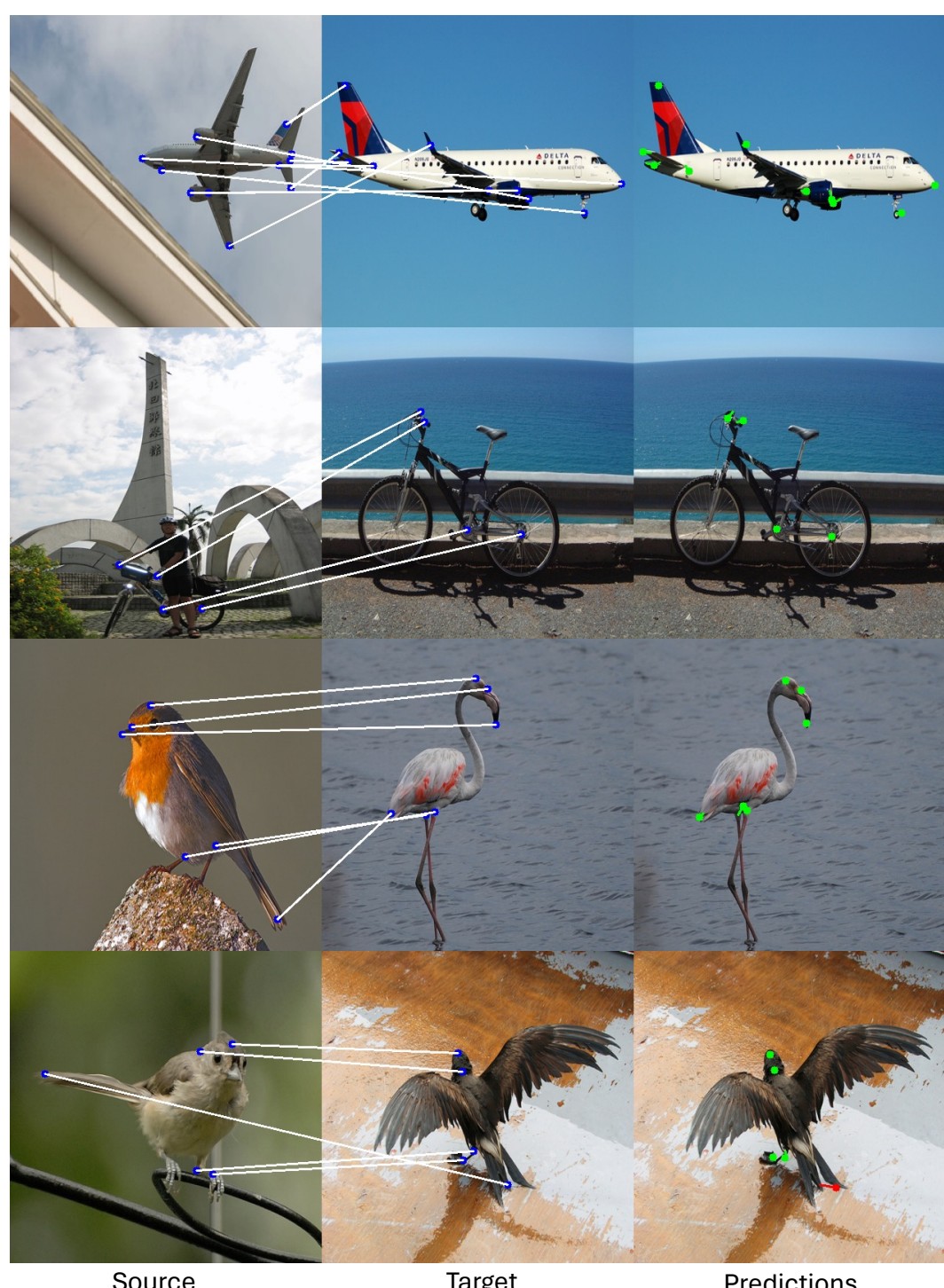

Figure 7: **Additional qualitative results of MambaMatcher**.

aggregation, exhibits low similarity at the GT position. Fig. 8 aims to provide a deeper insight into this phenomenon. In Fig. 8, the top-left images represent an image pair with a ground truth correspondence. The top-right image visualizes the output correlation map from MambaMatcher. Below these are $\{\mathbf{C}^i\}_0^{15}$, after the similarity-aware selective scan. As observed, some maps are completely noisy, while others accurately reflect the keypoint positions. This visualization helps

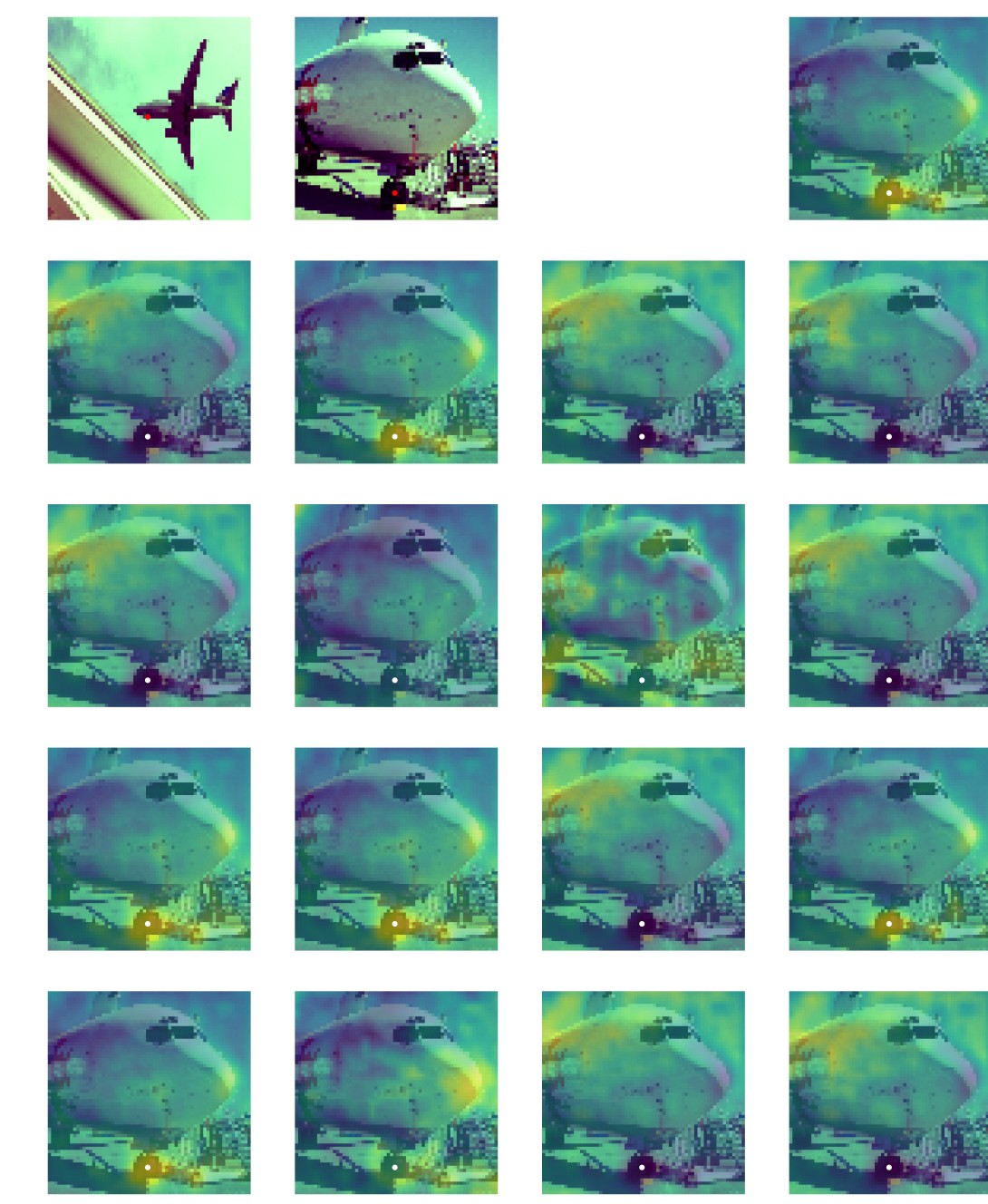

Figure 8: **Additional visualization of similarity-aware selective scan of MambaMatcher**. The top-left images represent an image pair with a ground truth correspondence. The top-right image visualizes the refined correlation map $\hat{\mathbf{C}}$ from MambaMatcher. Below these are $\{\mathbf{C}^i\}_0^{15}$, after the similarity-aware selective scan. As observed, some maps are completely noisy, while others accurately reflect the keypoint positions. This visualization helps illustrate that during the final prediction of $\hat{\mathbf{C}}$, the noisy maps are effectively disregarded, and the accurate maps are primarily weighted for aggregation, resulting in our final accurate correlation map.

illustrate that during the final prediction of $\hat{\mathbf{C}}$, the noisy maps are effectively disregarded, and the accurate maps are primarily weighted for aggregation, resulting in our final accurate correlation map.

