# OpenReview forum: "Selective State-Space Modeling of Correlation Maps for Semantic Correspondence"
_ICLR.cc/2025/Conference — Submitted to ICLR 2025_

### Official Review · Reviewer_aTzv · 2024-10-28

**Soundness:** 3
**Presentation:** 2
**Contribution:** 3
**Rating:** 5
**Confidence:** 5

**Summary:**

This paper proposes a new method called MambaMatcher, which applies selective state space models (SSMs) to semantic matching tasks and combines Mamba's linear complexity algorithm to efficiently process 4D correlation tensors. Compared with existing methods, MambaMatcher has improved the accuracy of keypoint localization without sacrificing computational efficiency, demonstrating significant performance improvements.

**Strengths:**

1. This paper introduces the first use of selective state-space models (SSMs) to efficiently model high-dimensional inter-image correlations.
2. This paper proposes a similarity-aware selective scan mechanism improves accuracy and efficiency in refining high-resolution correlations.
3. MambaMatcher integrates feature-metric and correlation-metric methods, achieving state-of-the-art performance with lower computational costs.

**Weaknesses:**

1. The specific description of the Correlation Aggregation via Similarity aware Selective Scan section is not clear enough.
2. There are some grammar errors in the paper ( e.g., Fig.4 Ground-truth correspondence and page 3, line 132).
3. Some formulas have commas or periods after them, while others do not. There is no uniformity in format ( e.g., Eq.10 and Eq.11).

**Questions:**

1. How does MambaMatcher balance the integration of feature-metric and correlation-metric methods?
2. Are there scenarios where this integration could present limitations or potential drawbacks?
3. Is there a significant difference in performance when processing images with different levels of complexity or size?

---

> ### Author Response · Authors · 2024-11-25
>
> We thank the reviewer for the constructive feedback. We provide the response to each point below.
>
> **W1. The specific description of the Correlation Aggregation via Similarity aware Selective Scan section is not clear enough.**
>
> The similarity-aware selective scan is a key part of our correlation aggregation process, designed to refine correspondences by effectively modeling inter-image relationships.
>
> We begin with the multi-level correlation map $\mathbf{C}$ of dimensions H×W×H×W×2L, where  $H$ and $W$ are the height and width of the feature maps, and $2L$ represents the concatenated multi-level features.
> To process this high-dimensional data efficiently, we reshape the 4D spatial correlation map into a 1D sequence: $\overline{\mathbf{C}} \in \mathbf{R}^{N\times 2L}$, where $N =$ HxWxHxW.
> This results in a sequence where each element corresponds to a potential match between pixels in the source and target images.
>
> We sort the sequence $\overline{\mathbf{C}}$  in descending order based on the similarity scores (correlation values). This sorting ensures that correspondences with higher similarity are processed first, allowing the model to prioritize the most reliable matches. By focusing on high-confidence matches early, we build a strong contextual foundation that helps disambiguate less certain correspondences encountered later.
>
> After sorting, we process the sequence using a state-space model implemented through the Mamba architecture, which efficiently captures sequential dependencies:
>
>
> 1. Projection of correlation sequence: The correlation sequence $\hat{\mathbf{C}}$ is projected to two intermediate features, $z$ and $x$.
> 2. Sequential modeling: The feature $x$ is processed via causal 1D convolution for sequential modeling in the state-space model, resulting in $x'$.
> 3. Computation of input-dependent parameters: Based on $x'$, we compute $\overline{\mathbf{B}}$ (how the input influences the state), $\mathbf{C}$ (how the current state translates to the output) and $\textrm{dt}$ (resolution of the input, i.e., discretization parameter). These parameters introduce input dependency, referred to as selectivity.
> 4. State evolution parameter: $\overline{\mathbf{A}}$ (How the current state evolves over time), ' is not input-dependent.
> 5. Selective scan mechanism: $\overline{\mathbf{A}}$, $\overline{\mathbf{B}}$,  $\hat{\mathbf{C}}$, and $\textrm{dt}$ are finally used for the similarity-aware selective scan mechanism to model the state-space effectively, outputting refined $x''$.
> 6. The final output is yielded by multiplication of $z$ and $x''$.
>
> In MambaMatcher, the multi-level similarity scores function as the 'states,' which are scanned in descending order of similarity scores. This approach ensures that the context primarily includes important and accurate cues, refining ambiguous correspondences effectively.
>
> We will incorporate these details into the manuscript as a pseudo-code to provide a more seamless and comprehensive explanation of our similarity-aware selective scan mechanism.
>
> **W2 and W3. Grammatical errors and inconsistencies in formulaes.**
>
> We apologize for these oversights. We will thoroughly review the paper to correct these mistakes. We will standardize the formatting throughout the manuscript to ensure consistency.
>
> **Q1. Balance of feature-metric and correlation-metric methods.**
>
> MambaMatcher harmoniously combines feature-metric and correlation-metric methodologies within a unified framework to enhance semantic correspondence tasks.
>
> 1. Feature-metric aggregation layers. We apply 2D convolutional layers to the feature maps extracted by the pre-trained backbone (i.e., DINOv2), enhancing the quality of the feature representations. By capturing local spatial context within each image, the 2D convolutions improve the distinctiveness and robustness of the features before computing correspondences.
>
> 2. Correlation-metric aggregation layers. We employ our similarity-aware selective scan mechanism, which processes the correlation maps by sorting the correlation scores in descending order of similarity. By prioritizing high-similarity correspondences, the model builds a reliable context that helps in refining ambiguous matches. The selective scan effectively models inter-image relationships by attending to the most significant correspondences first. This approach addresses cross-image ambiguities that cannot be resolved by feature refinement alone. As shown in Table 5 of our manuscript, our correlation-metric approach improves matching accuracy by refining the correspondences based on the similarity-aware sequence.
>
> Both the feature aggregation (feature-metric) layers and the correlation aggregation (correlation-metric) layers are trained jointly using a single loss function (Equation 8). This unified loss ensures that the learning of feature refinement and correlation refinement is coordinated.

---

> > ### Author Response · Authors · 2024-11-25
> >
> > **Q2. Potential limitations or drawbacks of integration.**
> >
> > Our identified potential drawbacks / limitations are as follows.
> >
> > 1. Memory constraints with ultra high-resolution images. While the efficiency of Mamba networks enables us to handle high-resolution feature maps, scaling to even larger image resolutions (e.g., over 1000 pixels in height or width) can lead to excessive memory consumption during training.
> >
> > 2. Challenges in hyperparameter search. The integration introduces additional hyperparameters for each aggregation layer, such as kernel sizes in the 2D convolutional layers and parameters within the similarity-aware selective scan mechanism.
> >
> > **Q3. Difference in performance w.r.t image resolution.**
> >
> > We have conducted experiments using image resolutions of 238, 420 (our default), and 840 to evaluate the impact of image size on the performance of MambaMatcher:
> >
> > | Image size | Feature size | correlation size | PCK @ 0.05 | PCK @ 0.10 | PCK @ 0.15 |
> > |---|---|---|---|---|---|
> > |238| $17^2$ | $17^4$ |  26.4 | 39.7 | 46.5 |
> > | 420 (Ours) | $30^2$ | $30^4$ | 61.6 | 77.8 | 84.3|
> > |840 | $60^2$ | $60^4$ | 64.2 | 78.4 | 85.2 |
> >
> > At the lower resolution of 238, we observed a noticeable decrease in performance compared to our default resolution of 420. At the higher resolution of 840, there was only a slight improvement over the default. We conjecture that the limited performance gain at the higher resolution is due to the lack of hyperparameter optimization specifically tailored for larger images. Since our model's hyperparameters were primarily tuned for the default resolution, adjusting them for higher resolutions could potentially yield more substantial improvements.
> >
> > Conducting extensive hyperparameter searches for different resolutions to optimize performance, or investigating methods to improve robustness against varying levels of image complexity—potentially through advanced feature representations or enhanced correlation modeling—are promising directions for future work.

---

> > > ### Author Response · Authors · 2024-12-02
> > >
> > > Dear Reviewer aTzv,
> > >
> > > Thank you once again for your valuable feedback on our paper. We have carefully addressed your questions and concerns in our responses. As the discussion period is coming to an end soon, we kindly encourage you to review our replies at your earliest convenience. We greatly appreciate your time and effort in reviewing our work.

---

### Official Review · Reviewer_Fxfi · 2024-11-02

**Soundness:** 3
**Presentation:** 3
**Contribution:** 3
**Rating:** 6
**Confidence:** 3

**Summary:**

MambaMatcher combines efficient high-dimensional correlation modeling with a selective scan mechanism, achieving superior performance on semantic correspondence benchmarks with lower computational costs.

While traditional feature-metric methods often fail to capture complex relationships between images, newer correlation-metric approaches address this but are hindered by the high computational cost of processing 4D correlation maps. MambaMatcher overcomes these issues by modeling high-dimensional correlations efficiently through selective state-space models (SSMs). Using a similarity-aware selective scan mechanism inspired by Mamba’s linear-complexity algorithm, MambaMatcher refines the 4D correlation tensor without compromising feature map resolution or receptive field. Experiments on standard semantic correspondence benchmarks demonstrate that MambaMatcher achieves state-of-the-art performance, capturing rich inter-image correlations while avoiding large input images and costly diffusion-based features.

**Strengths:**

1. Efficient Modeling of High-Dimensional Correlations: By using selective state-space models (SSMs), MambaMatcher models complex inter-image relationships effectively while avoiding the high computational cost typically associated with 4D correlation maps.

2. Novel Similarity-Aware Selective Scan Mechanism: The paper introduces a unique scan mechanism inspired by Mamba’s linear-complexity algorithm. This mechanism refines the 4D correlation tensor accurately, enabling high-resolution processing of inter-image correlations.

**Weaknesses:**

1. The content in Sec 4.2 MULTI-LEVEL CORRELATION COMPUTATION AND AGGREGATION is somewhat ambiguous. The overall process flow should be: Shape the correlation map from 4D to 1D -> order -> reorder -> reshape  the correlation map  from 1D to 4D. Is the sorting performed along the $H \times W \times H \times W$ dimension or along the $2L$ dimension? If it is along the $H \times W \times H \times W$ dimension, is the sorting directly based on similarity in descending order? How are the potential spatial relationships between correlation elements encoded?

2. Similarity-aware Selective Scanning: Mamba relies on its inherent order to capture the relative positional relationships between sequence tokens. When  correlation elements are sorted by similarity before scanning, does this mean  no longer need to consider the spatial locations of these elements?

3. Parameters of Feature Aggregation: Line 215 states that "The feature aggregators share the same weights across all levels to maintain efficiency." However, Lines 320-322 describe the feature aggregation layer as consisting of two layers of 2D convolution with a kernel size of 5, with output channel dimensions of 64 and 14, respectively, and a ReLU activation function in between. This implies that feature aggregation includes four 2D convolutional layers and two ReLU activations. However, in Table 6, feature aggregation is reported to have 42.5M parameters, which appears inconsistent. Please provide a more detailed explanation.

**Questions:**

1. Ambiguity in Feature and Correlation Aggregations. Section 4.1 introduces Multi-level Feature Aggregation using a lightweight 2D convolution network, while Section 4.2 proposes MULTI-LEVEL CORRELATION COMPUTATION AND AGGREGATION using Mamba. Both aggregation modules require training, but throughout the text, the authors often refer to “the aggregation” without specifying which aggregation they mean. This lack of clarity can easily lead to confusion, as in Lines 251-352: “We freeze the visual feature extractor during training to focus on learning the aggregation layers.” Here, does “the aggregation layers” refer to the feature aggregation in Section 4.1, the correlation aggregation in Section 4.2, or both?

2.Effects of selective scan order. Table 5 indicates that the relative order of correlation elements significantly impacts performance, with ascending order notably reducing effectiveness compared to descending order. This may be because, in causal inference, later tokens pass information to all preceding tokens. Placing high-similarity tokens at the beginning can help reduce interference from other correlation elements. The paper briefly claims that "Early processing of strong matches helps resolve ambiguities in these regions," in Sec. 4.2, but a more in-depth analysis is needed.

---

> ### Author Response · Authors · 2024-11-25
>
> We thank the reviewer for the constructive feedback. We provide the response to each point below.
>
> **W1: The content in Sec 4.2 MULTI-LEVEL CORRELATION COMPUTATION AND AGGREGATION is somewhat ambiguous.**
>
> We apologize for any ambiguity in our explanation and appreciate the opportunity to clarify the process. The overall process involves reshaping the 4D correlation map into a 1D  correlation sequence, sorting it, processing it, and then reshaping it back to its original dimensions:
> 1. Reshaping to 1D. We reshape the 4D correlation map of dimensions HxWxHxWx2L into a 1D sequence of length N = H×W×H×W, where each element represents a correlation score between a pair of spatial locations from the source and target images.
> 2. Sorting based on similarity. The sorting is performed along this 1D sequence of length N, directly based on the similarity scores from the final-level correlation map in descending order. This means that correspondences with higher similarity scores are prioritized early in the sequence.
> 3. Processing the sequence. After sorting, we apply our similarity-aware selective scan mechanism to the sequence. This allows the model to build context starting from the most reliable and accurate correspondences, effectively disambiguating the scores.
> 4. Reshaping back to 4D: The processed sequence is then reshaped back into the original 4D correlation map dimensions for further processing and final correspondence prediction.
>
> **W1 and W2: Encoding of spatial relationships.**
>
> Currently, we do not explicitly encode the potential spatial relationships between correlation elements during the sorting and processing steps. While spatial relationships are important, our primary goal is to resolve ambiguities by focusing on the most significant correspondences first. Prioritizing high-similarity scores enables the model to establish a strong contextual foundation.
>
> We explored several methods to incorporate spatial relationships:
> 1. Alternative scanning orders: We experimented with (bi-)directional scanning and Z-order curves to preserve spatial proximity within the sequence. However, as shown in Table 5, these approaches did not outperform our similarity-aware selective scan.
> 2. 4D sinusoidal positional encoding: To further analyze the effect of encoding spatial relationships, we also attempted to encode spatial information using 4D sinusoidal positional encodings. The results evaluated on SPair-71k are as follows:
>
> |Method | PCK @ 0.05 | PCK @ 0.10| PCK@0.015|
> |---|---|---|---|
> |Ours| 61.6 | 77.8 | 84.3|
> |Ours with P.E.| 61.2 | 77.8 | 84.2|
>
> It can be seen that the difference in results are negligible.
>
> We acknowledge that effectively encoding spatial relationships within the correlation aggregation process remains an important and promising area for future research. Investigating more sophisticated methods to integrate spatial context could potentially enhance the model's ability to capture inter-image relationships. However, existing conventional methods for positional embedding (sinusoidal embedding, or directional scans in Mamba-related work) show to perform worse than our current scheme of similarity-aware selcetive scan.
>
> **W3: Parameters of Feature Aggregation.**
>
> Thank you for bringing attention to the inconsistency regarding the parameters of the Feature Aggregation module. We apologize for any confusion caused by this oversight in our manuscript. In Lines 320-322, we mistakenly stated that the feature aggregation layer consists of two 2D convolutional layers with a kernel size of 5 and output channel dimensions of 64 and 14, respectively, with a ReLU activation in between.
>
> The actual architecture of our feature aggregation layer consists of two 2D convolutional layers with output channel dimensions at 4 × 768 (i.e., 3072) for the first convolutional layer and 768 for the second, with a ReLU activation function in between. This is based on the channel dimension of features extracted from the DINOv2 backbone, which is 768. Also, the the current parameter count was wrongly calculated at kernel size of 3; at our current setting of kernel size of 5, the actual parameter count amounts up to 118M. The initial miscalculation was due to an error in recording the kernel sizes and channel dimensions.
>
> We will update Lines 320-322 to accurately reflect the correct kernel sizes and output channel dimensions of the feature aggregation layers. Specifically, we will correct the kernel sizes to 3 and the output channels to 3072 and 768, respectively. We will revise Table 6 to reflect the accurate parameter count of approximately 118 million for the feature aggregation module. We will ensure consistency throughout the manuscript regarding the architecture and parameter details to prevent any further confusion. We sincerely apologize for the confusion caused by these inaccuracies.

---

> ### Author Response · Authors · 2024-11-25
>
> **Q1. Ambiguity in Feature and Correlation Aggregations.**
>
> We apologize for any confusion this may have caused. You are correct that both the Multi-Level Feature Aggregation described in Section 4.1 and the Multi-Level Correlation Computation and Aggregation using Mamba in Section 4.2 involve aggregation modules that require training.
> In the sentence you referenced (Lines 251-252): "We freeze the visual feature extractor during training to focus on learning the aggregation layers," the term "aggregation layers" refers to both the feature aggregation layers **and** the correlation aggregation layers. Specifically, after extracting features using the pre-trained DINO backbone (which we keep frozen), we apply 2D convolutional layers to enhance and aggregate the features. These layers are trainable and are crucial for improving the quality of the feature representations. We then compute the multi-level correlation maps and process them using our similarity-aware selective scan mechanism (the Mamba module). This component is also trainable and is responsible for refining the correspondences between images.
>
> We will revise the manuscript to explicitly specify whether we are referring to feature aggregation, correlation aggregation, or both.
>
> **Q2: Effects of selective scan order.**
>
> You are correct that the relative order of correlation elements significantly impacts performance, as shown in Table 5, where the descending order outperforms the ascending order.
>
> The primary reason for this performance difference lies in how the context is built during the correlation aggregation process. In our method, the context is constructed selectively within the correlation aggregation layers based on Mamba. By sorting the correlation elements in descending order of similarity scores, we prioritize high-similarity correspondences early in the sequence. This approach allows the model to initially establish a strong and reliable contextual foundation based on accurate matches.
>
> Processing high-similarity tokens first helps to disambiguate ambiguous matches encountered later. Since the initial context is built upon the most reliable information, it reduces interference from less certain correspondences. Conversely, processing low-similarity correspondences first (ascending order) can introduce noise into the context, making it more challenging to resolve ambiguities in subsequent matches.
>
> We acknowledge that our brief mention in Section 4.2—that "early processing of strong matches helps resolve ambiguities in these regions"—may not provide sufficient depth.  Specifically, we will discuss how the selective scan order influences context building and the disambiguation of matches during inference. Additionally, we will reference Figure 5 to illustrate how the descending order scheme improves matching accuracy compared to the ascending order. This figure demonstrates that prioritizing high-similarity correspondences results in more accurate and reliable matching outcomes, thereby validating our approach.

---

> > ### Author Response · Authors · 2024-12-02
> >
> > Dear Reviewer Fxfi,
> >
> > Thank you once again for your valuable feedback on our paper. We have carefully addressed your questions and concerns in our responses. As the discussion period is coming to an end soon, we kindly encourage you to review our replies at your earliest convenience. We greatly appreciate your time and effort in reviewing our work.

---

> > > ### Comment · Reviewer_Fxfi · 2024-12-02
> > >
> > > Thank you to the authors for thoroughly addressing my questions and concerns, especially on W1 and W2 (Encoding of spatial relationships), W3 (Parameters of Feature Aggregation), and Q2 (Effects of selective scan order). However, considering the feedback from other reviewers, I prefer to maintain my initial score. Thanks.

---

### Official Review · Reviewer_RJWE · 2024-11-04

**Soundness:** 3
**Presentation:** 2
**Contribution:** 3
**Rating:** 5
**Confidence:** 4

**Summary:**

The paper introduces MambaMatcher for high-dimensional correlation modeling using selective state-space models, uniquely treating multi-level correlation scores as states to capture inter-image correlations. A key feature is the similarity-aware selective scan mechanism, enabling efficient, high-resolution correlation extraction. MambaMatcher combines feature-metric and correlation-metric approaches in a unified pipeline without compromising feature map resolution or receptive field. Extensive experiments show state-of-the-art performance on semantic correspondence benchmarks, outperforming diffusion-based methods with lower computational cost, making MambaMatcher an effective and efficient solution.

**Strengths:**

1. The paper is well-written and easy to understand.
2. The paper presents extensive experiments, demonstrating state-of-the-art performance on benchmark datasets.
3. The paper provides a thorough comparison of different feature aggregation and correlation aggregation structures, offering valuable insights into how these configurations affect model performance.

**Weaknesses:**

1. The motivation is unclear. While the authors claim that their method addresses the challenge of capturing inter-image correlations, they do not clearly define what constitutes an inter-image correlation problem or why it is important. A more detailed motivation and definition would help clarify the significance of this aspect for the reader.
2. The novelty of the approach appears limited, as the correlation structure used in this method closely resembles that of CAT++, raising concerns about its originality.
3. The visualizations in Figure 5 and Figure 3 are not easy to understand and could benefit from clearer explanations.

**Questions:**

1. Time and Space complexity should be analyzed for the proposed algorithms, and also the key psuedo-code should be presented in this paper.
2. In the context of the paper, could you clarify what is meant by 'inter-image relationships'? Specifically, how does this differ from identifying semantic similarities across local pixels within an image, and why is it particularly challenging for feature-metric methods to capture these relationships?
3. The results in Table 1 have inconsistent resolutions, making it necessary to compare them at the same resolution to fully demonstrate the effectiveness of the proposed method.
4. Figure 5 is somewhat difficult to interpret, as it’s unclear what the red and green points represent. Providing a clearer legend or explanation for these colors would enhance the reader’s understanding of the figure’s purpose and improve overall clarity.

---

> ### Author Response · Authors · 2024-11-25
>
> We thank the reviewer for the constructive feedback. We provide the response to each point below.
>
> **W1. The motivation is unclear. A more detailed motivation and definition would help clarify the significance of this aspect for the reader.**
>
> Relying solely on strong feature representations is suboptimal because intra-image features cannot resolve cross-image ambiguities that occur when similar structures or textures appear in different regions of the images. This limitation is why prior works like NCNet, CHMNet, TransforMatcher, CATs, SuperGlue, and LoFTR incorporate mechanisms to attend to both images, effectively capturing inter-image correlations to reduce these ambiguities.
>
> Our method addresses this challenge by modeling the multi-level correlation scores between images as 'states' in a state-space model. By attending to the strongest correspondences first through our similarity-aware selective scan, we build a reliable foundation for our state-space mechanism, which then refines ambiguous matches more effectively. Unlike conventional image matching tasks where the goal is to find as many correct matches as possible, semantic correspondence requires predicting precise target points for fixed source keypoints. This means we cannot simply discard ambiguous matches; instead, we must rectify them to achieve accurate correspondences.
>
> We will revise the manuscript to clearly define what constitutes an inter-image correlation problem and explain its significance in semantic correspondence.
>
> **W2. The novelty of the approach appears limited, as the correlation structure used in this method closely resembles that of CAT++.**
>
> We appreciate the opportunity to clarify the distinctions between our method and CATs++, addressing your concerns about originality.
> Firstly, while both our method and CATs++ utilize correlation maps, the structures and processing techniques are fundamentally different:
>
> - Correlation Structure Differences: CATs++ computes a multi-level correlation map of shape (H×W×H×W)×L and then concatenates it with projected multi-level features of shape (H×W)×p×L, forming a reshaped structure of (H×W)×(H×W+p)×L. This results in a combined representation that includes both correlation and feature information.
> - Our Approach: In contrast, we compute a multi-level correlation map of shape (H×W×H×W)×2L without concatenating additional projected features. We rely solely on the multi-level correlation map to capture rich matching information, which simplifies the model and focuses on the correlation relationships directly.
>
> Moreover, our method outperforms CATs++ while relying just on the multi-level correlation map, demonstrating the effectiveness of our approach without the need for concatenating features.
> We acknowledge that multi-level correlation maps are not novel per se and have been employed in prior works such as NCNet and CHMNet (single-level correlation maps) and TransforMatcher and HCCNet (multi-level correlation maps). However, the novelty of MambaMatcher lies not in the correlation structure itself but in how we process and model these correlations:
>
> - Harmonious Integration of Feature-Metric and Correlation-Metric Approaches: We uniquely integrate feature refinement through 2D convolutions (the feature-metric approach) with our similarity-aware selective scan mechanism (the correlation-metric approach). This combination leverages the strengths of both methodologies to enhance overall performance, as shown in Tables 4 and 5.
> - Novel Conceptual Modeling: We introduce a conceptual innovation by treating multi-level correlation scores as 'states' in an efficient state-space model. By interpreting the correlation map as a sequence in a similarity-aware manner, we refine correspondences more effectively. This is achieved through our similarity-aware selective scan, which prioritizes high-similarity correspondences by sorting the correlation sequences in descending order of similarity scores (mentioned in Line 248).
>
> Our approach addresses limitations in previous methods:
>
> - Efficiency without Sacrificing Performance: Unlike methods that either use high-order convolutions with static weights (e.g., PWarpCNet) or reshape the correlation map at the cost of structural information (e.g., TransforMatcher), our method efficiently attends globally to the full 4D space without reducing correlation size or losing performance.
> - Effective Processing of Large Correlation Maps: As detailed in our response to Reviewer Xgjj (Comment 2.1) and Table 17 of the appendix, MambaMatcher can handle large correlation maps derived from higher-resolution feature maps (e.g., 60×60 features leading to $60^4$ elements) efficiently, which is challenging for existing methods.
>
> In summary, while we build upon existing structures, our key contributions are the innovative processing and modeling of correlation maps and the harmonious integration of feature-metric and correlation-metric approaches.

---

> ### Author Response · Authors · 2024-11-25
>
> **W3: The visualizations in Figure 5 and Figure 3 are not easy to understand.**
>
> Figure 3 aims to illustrate the effect of our similarity-aware selective scan mechanism:
>
> - Visualization Details: The leftmost two columns display the source and target images, with the source keypoint marked by a red dot and its corresponding ground truth target keypoint also indicated by a red dot in the target image.
>
> - Correlation Map Selection: Given that we have H×W multi-level correspondence maps per image pair (each of size H×W), visualizing the entire set is impractical. Therefore, we focus on the 11th correlation map, denoted as $\mathbf{C}^{11}$, for clarity.
>
> - Pre- and Post-Aggregation Comparison: The third column shows the pre-aggregation correlation map, which is noisy with high similarity scores across multiple regions. After applying our similarity-aware selective scan (post-aggregation), the fourth column reveals a refined correlation map where the similarity scores are more concentrated but may still not precisely highlight the ground truth location.
>
> - Final Prediction: The last column presents the single-level correlation map $\hat{\mathbf{C}}$ after the final transformation from the refined multi-level correlation map. Here, the target keypoint is accurately localized, demonstrating how our method effectively aggregates information to enhance precision.
>
> We discuss this process further in Appendix L, showing that depending on the level in the multi-level correlation map, individual maps can be noisy or accurate. Our aggregation method prioritizes the accurate maps, resulting in a final correlation map that reliably predicts the correct correspondences.
>
> Figure 5 displays the results of our semantic matching:
> - Ground Truth Correspondences: The first two columns show the source and target images with the ground truth keypoints.
> - Predicted Keypoints: The rightmost three columns illustrate the predicted target keypoints. Green circles represent accurate predictions within a certain threshold, while red circles indicate inaccuracies beyond that threshold.
> - Error Visualization: Lines connecting the predicted and ground truth keypoints depict the deviation—the shorter the line, the more accurate the prediction.
> We acknowledge that our initial explanations may not have been sufficiently detailed. We will update the manuscript to provide clearer and more straightforward descriptions of these figures.
>
> **Q1. Time and Space complexity should be analyzed for the proposed algorithms.**
>
> We guide the reviewer to section E of the supplementary, where we have included the time complexity analyses for ours, and other existing schemes. For the space complexity of our similarity-aware selective scan, it can be represented as $O(C)$, where $C$ reflects the fixed size of the state-space model latent state. Note that this is a significantly compressed latent state, benefitting from the input-dependent modeling.
>
> To clarify the process, we will include the following detailed explanation corresponding to Figure 2:
>
> 1. Projection of correlation sequence: The correlation sequence $\hat{\mathbf{C}}$ is projected to two intermediate features, $z$ and $x$.
> 2. Sequential modeling: The feature $x$ is processed via causal 1D convolution for sequential modeling in the state-space model, resulting in $x'$.
> 3. Computation of input-dependent parameters: Based on $x'$, we compute $\overline{\mathbf{B}}$ (how the input influences the state), $\mathbf{C}$ (how the current state translates to the output) and $\textrm{dt}$ (resolution of the input, i.e., discretization parameter). These parameters introduce input dependency, referred to as selectivity.
> 4. State evolution parameter: $\overline{\mathbf{A}}$ (How the current state evolves over time), ' is not input-dependent.
> 5. Selective scan mechanism: $\overline{\mathbf{A}}$, $\overline{\mathbf{B}}$,  $\hat{\mathbf{C}}$, and $\textrm{dt}$ are finally used for the similarity-aware selective scan mechanism to model the state-space effectively, outputting refined $x''$.
> 6. The final output is yielded by multiplication of $z$ and $x''$.
>
> In MambaMatcher, the multi-level similarity scores function as the 'states,' which are scanned in descending order of similarity scores. This approach ensures that the context primarily includes important and accurate cues, refining ambiguous correspondences effectively.
>
> We will incorporate these details into the manuscript as a pseudo-code to provide a more seamless and comprehensive explanation of our similarity-aware selective scan mechanism.

---

> ### Author Response · Authors · 2024-11-25
>
> **Q2. In the context of the paper, could you clarify what is meant by 'inter-image relationships'?**
>
> In the context of our paper, "inter-image relationships" refer to the correspondences and dependencies between features across two different images. This involves understanding how specific pixels or regions in one image relate semantically and spatially to pixels or regions in another image. This is distinct from identifying semantic similarities across local pixels within a single image (intra-image analysis), which focuses solely on the relationships and structures within that image.
>
> Feature-metric methods enhance and extract robust features within individual images, effectively capturing intra-image relationships. While these methods are effective—as evidenced by the results in Table 4—they may not fully resolve ambiguities that arise when matching features across images. This is because there can be repeated or similar patterns in both images (e.g., eyes, ears, windows), leading to multiple potential matches for a single feature. Homogeneous regions may also present ambiguities due to lack of distinctive features. Features extracted independently from each image might not capture the necessary contextual information to disambiguate these potential matches. This can result in similarity scores that are neither high enough to confirm a true match (inlier) nor low enough to exclude a false one (outlier).
>
> To address these challenges, it is crucial to model the inter-image relationships explicitly. By analyzing how features from one image correlate with features in another, we can refine similarity scores and reduce ambiguities. Our method integrates both feature-metric and correlation-metric approaches. While we enhance features within each image, we also model the inter-image relationships through our similarity-aware selective scan mechanism. This dual strategy helps in refining scores and effectively disambiguating matches.
>
> **Q3: The results in Table 1 have inconsistent resolutions, making it necessary to compare them at the same resolution to fully demonstrate the effectiveness of the proposed method.**
>
> Due to computational limitations, it is challenging to re-run all the baseline methods across different resolutions. Many existing semantic matching methods and their architectures are specifically tailored to operate optimally at certain image resolutions. Adjusting these methods to work effectively at a uniform resolution often requires extensive hyperparameter tuning, which can be prohibitively time-consuming and resource-intensive.
>
> We have provided results for varying image sizes—specifically resolutions of 238, 420, and 840—to offer insight into how different resolutions impact performance.
>
> | Image size | Feature size | correlation size | PCK @ 0.05 | PCK @ 0.10 | PCK @ 0.15 |
> |---|---|---|---|---|---|
> |238| $17^2$ | $17^4$ |  26.4 | 39.7 | 46.5 |
> | 420 (Ours) | $30^2$ | $30^4$ | 61.6 | 77.8 | 84.3|
> |840 | $60^2$ | $60^4$ | 64.2 | 78.4 | 85.2 |
>
> While the performance varies with the image size, and ours underperforms many methods at the lower image dimensions of 238x238, we conjecture this is because the hyperparameters in our pipeline e.g., convolutional kernel size or gaussian kernel ($\mathbf{G}$) size, are relatively sensitive to the input image dimensions. We hypothesize that careful tuning of these hyperparameters will maximize the performance of MambaMatcher across different image dimensions.
>
> **Q4: Figure 5 is somewhat difficult to interpret.**
>
> We acknowledge that it may be difficult to interpret, and we apologize for any confusion. We have provided a detailed explanation of Figure 5 in our response to W3. In the revised manuscript, we will incorporate these explanations directly into the text and enhance the visual presentation of the figure to improve clarity and ease of understanding.

---

> > ### Author Response · Authors · 2024-12-02
> >
> > Dear Reviewer RJWE,
> >
> > Thank you once again for your valuable feedback on our paper. We have carefully addressed your questions and concerns in our responses. As the discussion period is coming to an end soon, we kindly encourage you to review our replies at your earliest convenience. We greatly appreciate your time and effort in reviewing our work.

---

### Official Review · Reviewer_Xgjj · 2024-11-05

**Soundness:** 3
**Presentation:** 4
**Contribution:** 2
**Rating:** 6
**Confidence:** 4

**Summary:**

This paper proposes MambaMatcher, a Mamba-based approach to establish semantic correspondences. MambaMatcher attempts to model inter-image correlations by treating multi-level correlation scores at each position in the correlation map as a state in a state-space model. Specifically, the multi-level features obtained by the frozen extractor are first enhanced through feature aggregation. The enhanced features are used to construct a multi-level correlation map. Then, the correlation map is refined with the proposed similarity-aware selective scan mechanism. Experimental results demonstrate that MambaMatcher achieves a good balance between accuracy and efficiency in establishing semantic correspondences.

**Strengths:**

1. Mamba is a potential architecture to balance the accuracy and efficiency of establishing semantic correspondences. It is meaningful to explore how to apply the Mamba structure in this task.
2. The organization and presentation of this paper are clear.

**Weaknesses:**

**1. The core design of MambaMatcher, i.e., treating the correlation scores at each position in the correlation map as a state in a state-space model, seems to provide only a minor benefit.**

According to the results in Table 4 and Table 5, the combination of a DINOv2 feature extractor, a 2D-Conv_{k=5} feature aggregation and the existing correlation aggregation processes (based on 4D-Conv or FastFormer) provide a similar accuracy compared with the proposed MambaMatcher. This phenomenon indicates that the selection of feature extractors and the design of single-image feature aggregation may be more significant problems in establishing semantic correspondences. Replacing the existing 4D-Conv or FastFormer with the Mamba structure in the correlation aggregation process just gives a small improvement. Such a performance decreases the significance of the motivation and the core design in this paper.

**2. Some claims in this paper are not discussed clearly.**

2.1. In Line 48, the authors claim that one problem of the existing correlation-metric approaches is that they “severely limit the feature map resolution”. However, the proposed MambaMatcher only processes the 30x30 feature map. Could we consider the 30x30 size as a large resolution? Maybe more experiments are required to validate MambaMatcher’s superiority in handling larger feature maps.

2.2. In Line 69, the authors claim that “MambaMatcher seamlessly integrates feature-metric and correlation-metric approaches into a unified pipeline.” It indicates that some parts of MambaMatcher are feature-metric-based while some are correlation-metric-based. However, there is no discussion to clarify such a claim.

**3. The introduction of the proposed “similarity-aware selective scan mechanism” is not clear enough.**

The first reason of this problem is that Figure 2 is not cited in the main text. Besides, some equations may make the process of “similarity-aware selective scan mechanism” clearer.

**Questions:**

Please provide more discussions and experimental results to address the above weaknesses.

-------------After the Discussion Period---------------

Thank the authors for the clear and detailed responses to my questions. As mentioned in the strengths I concluded, I consider this work as a meaningful exploration. I also agree with the authors that MambaMatcher can complement some existing components. However, my concern in W1 is mainly about the significance of this work, which is not addressed well in the rebuttal. According to the responses, the core Mamba-based module provides only modest improvements compared with some other parts like feature extractors and single-image feature aggregations. These results indicate that the significance of this work is relatively small. Considering both such strengths and weaknesses, I prefer to maintain my initial score. Thanks.

Moreover, I can understand the responses for W2.1. However, in my opinion, the statements about “high resolution” in the manuscript are confusing because the authors did not clarify that the resolution is discussed for the 4D correlation map rather than the original 2D feature map.

---

> ### Author Response · Authors · 2024-11-25
>
> We thank the reviewer for the constructive feedback. We provide the response to each point below.
>
> **1. The core design of MambaMatcher seems to provide only a minor benefit.**
>
> We agree that strong feature extractors and single-image feature aggregation are critical in establishing semantic correspondences, as the correspondences are computed from these features. This is why performance varies significantly with different feature-metric approaches. Our aim with MambaMatcher is to **complement** these components by refining the correlation aggregation process to maximize the potential of the features.
>
> While the improvements from our correlation-metric scheme may appear modest compared to those from feature-metric enhancements, MambaMatcher consistently yields the best results among evaluated methods, as shown in Tables 4 and 5. Notably, certain existing correlation-metric schemes (e.g., 4D convolution with kernel size > 1) can degrade performance, underscoring the need for a more effective approach. Our similarity-aware selective scan addresses this by efficiently refining correspondences, resulting in superior performance and efficiency. As detailed in Table 6, our method is not only more accurate but also faster than existing correlation aggregation processes.
>
> We acknowledge that there is significant room for advancement in correlation-metric schemes, presenting a promising future direction with substantial potential. By integrating MambaMatcher harmoniously with strong feature extractors, we enhance the overall semantic matching pipeline without conflicting with their strengths. We believe our work contributes meaningful improvements and opens avenues for further research in this area.
>
> **2.1. Could we consider the 30x30 size as a large resolution? Maybe more experiments are required to validate MambaMatcher’s superiority in handling larger feature maps.**
>
> You are correct that a 30×30 feature map may not be considered large in absolute terms. However, the limitation arises not from the feature map size itself but from the subsequent size of the 4D correlation map generated during the matching process.
>
> Even with a 30×30 feature map, the resulting correlation map expands to $30^4=810,000$ elements. This exponential growth poses significant computational and memory challenges. Existing correlation-metric approaches often struggle with such large correlation maps and resort to compromises:
> - Higher-order convolutions: Methods like PWarpCNet / NCNet use high-order convolutions with static weights, which provide only local context and limit the ability to capture global relationships.
> - Reshaping techniques: Methods like CATs  reshape the correlation map to reduce token length (e.g., from B×H×W×H×W×C to (B×HW)×HW×C), but this can limit performance, as the model cannot attend to the global correlation space simultaneously.
> - Efficient, but less performant architectures: Architectures like TransforMatcher aim for efficiency but may sacrifice accuracy due to less effective handling of the full correlation space (i.e., using FastFormers, an additive attention mechanism for efficiency).
>
> Our proposed MambaMatcher overcomes these limitations by enabling efficient global attention over the full 4D correlation space without reducing the correlation size or sacrificing performance. As demonstrated in Table 17 of our appendix, MambaMatcher can handle larger feature maps—such as 50×50, leading to a correlation map of $50^4$ elements—while maintaining efficient computational overhead during inference. This capability contrasts with methods like the vanilla transformer, which encounter out-of-memory issues even at a batch size of one due to their quadratic complexity during training.
>
> We present additional results at varying image resolutions on the SPair-71k dataset below:
>
> | Image size | Feature size | correlation size | PCK @ 0.05 | PCK @ 0.10 | PCK @ 0.15 |
> |---|---|---|---|---|---|
> |238| $17^2$ | $17^4$ |  26.4 | 39.7 | 46.5 |
> | 420 (Ours) | $30^2$ | $30^4$ | 61.6 | 77.8 | 84.3|
> |840 | $60^2$ | $60^4$ | 64.2 | 78.4 | 85.2 |
>
> While the performance varies with the image size, and ours underperforms many methods at the lower image dimensions of 238x238, we conjecture this is because the hyperparameters in our pipeline e.g., convolutional kernel size or gaussian kernel ($\mathbf{G}$) size, are relatively sensitive to the input image dimensions. We hypothesize that careful tuning of these hyperparameters will maximize the performance of MambaMatcher across different image dimensions.

---

> ### Author Response · Authors · 2024-11-25
>
> **2.2.Line 69 indicates that some parts of MambaMatcher are feature-metric-based while some are correlation-metric-based. However, there is no discussion to clarify such a claim.**
>
> Thank you for highlighting the need for clarification regarding our claim in Line 69 that "MambaMatcher seamlessly integrates feature-metric and correlation-metric approaches into a unified pipeline." We acknowledge that this integration may not have been made sufficiently clear in the manuscript.
>
> In our work, MambaMatcher indeed combines both feature-metric and correlation-metric methodologies:
>
> - Feature-Metric Approach: We enhance the features extracted from images using 2D convolutions. These convolutional layers improve the quality of the feature representations before computing correspondences. This component is analyzed in Table 4, where we compare different feature-metric approaches and demonstrate their impact on performance.
>
> - Correlation-Metric Approach: Our similarity-aware Mamba scan operates on the correlation map to refine correspondences between images. By effectively processing the correlation map, we improve matching accuracy. This method is detailed and compared in Table 5, showcasing the benefits of our correlation-metric approach over existing techniques.
>
> By integrating these two components, MambaMatcher leverages the strengths of both approaches to enhance semantic correspondence tasks. While we discussed these methodologies in the Related Work section under "Feature-Metric Approaches for Semantic Correspondence" and "Correlation-Metric Approaches for Semantic Correspondence," we realize that the unified nature of our pipeline may not have been explicitly conveyed.
> We will revise the manuscript to make this integration clearer and more straightforward.
>
> **3. The introduction of the proposed “similarity-aware selective scan mechanism” is not clear enough.**
>
> We apologize for not citing Figure 2 in the main text; we will address this oversight in our revision to ensure the figure effectively supports our explanation.
>
> The "similarity-aware" aspect of our mechanism comes from sorting the correlation sequence in descending order of similarity scores, as mentioned in Line 248. This sorting prioritizes correspondences with higher similarity, allowing the model to build context based on the most significant matches first. After sorting, the mechanism proceeds with the selective scan process, as illustrated in our Preliminary section (Equations 1 to 3).
>
> To clarify the process, we will include the following detailed explanation corresponding to Figure 2:
>
> 1. Projection of correlation sequence: The correlation sequence $\hat{\mathbf{C}}$ is projected to two intermediate features, $z$ and $x$.
> 2. Sequential modeling: The feature $x$ is processed via causal 1D convolution for sequential modeling in the state-space model, resulting in $x'$.
> 3. Computation of input-dependent parameters: Based on $x'$, we compute $\overline{\mathbf{B}}$ (how the input influences the state), $\mathbf{C}$ (how the current state translates to the output) and $\textrm{dt}$ (resolution of the input, i.e., discretization parameter). These parameters introduce input dependency, referred to as selectivity.
> 4. State evolution parameter: $\overline{\mathbf{A}}$ (How the current state evolves over time), ' is not input-dependent.
> 5. Selective scan mechanism: $\overline{\mathbf{A}}$, $\overline{\mathbf{B}}$,  $\hat{\mathbf{C}}$, and $\textrm{dt}$ are finally used for the similarity-aware selective scan mechanism to model the state-space effectively, outputting refined $x''$.
> 6. The final output is yielded by multiplication of $z$ and $x''$.
>
> In MambaMatcher, the multi-level similarity scores function as the 'states,' which are scanned in descending order of similarity scores. This approach ensures that the context primarily includes important and accurate cues, refining ambiguous correspondences effectively.
>
> We will incorporate these details into the manuscript to provide a more seamless and comprehensive explanation of our similarity-aware selective scan mechanism.

---

> > ### Author Response · Authors · 2024-12-02
> >
> > Dear Reviewer Xgjj,
> >
> > Thank you once again for your valuable feedback on our paper. We have carefully addressed your questions and concerns in our responses. As the discussion period is coming to an end soon, we kindly encourage you to review our replies at your earliest convenience. We greatly appreciate your time and effort in reviewing our work.

---

### Meta-Review · Area_Chair_RACa · 2024-12-23

**Metareview:**

This paper presents MambaMatcher, a method using Mamba to establish semantic correspondences. By modeling inter-image correlations through multi-level correlation scores and a state-space model, MambaMatcher enhances features via aggregation, constructs a correlation map, and refines it with a similarity-aware selective scan. Experimental results show that MambaMatcher effectively balances accuracy and efficiency in establishing semantic correspondences.

**Additional Comments On Reviewer Discussion:**

After the rebuttal, the paper receives 2xboarderline reject and 2xboarderline accept. The major concerns are the novelty, experimental results, and the presentation. Considering all the reviews and the discussion, the AC considers that the paper is not ready at the moment, and would encourage the authors to improve it for a future submission considering the feedbacks from the reviewers.

---

### Decision · Program_Chairs · 2025-01-22

Reject